# Robust Barycenter Estimation using Semi-Unbalanced Neural Optimal Transport

**Milena Gazdieva**[*]
Skolkovo Institute of Science and Technology
Artificial Intelligence Research Institute
Moscow, Russia
milena.gazdieva@skoltech.ru

**Jaemoo Choi**[*]
Georgia Institute of Technology
Atlanta, GA, USA
jchoi843@gatech.edu

**Alexander Kolesov**
Skolkovo Institute of Science and Technology
Artificial Intelligence Research Institute
Moscow, Russia
a.kolesov@skoltech.ru

**Jaewoong Choi**
Sungkyunkwan University
Seoul, Korea
jaewoongchoi@skku.edu

**Petr Mokrov**
Skolkovo Institute of Science and Technology
Moscow, Russia
p.mokrov@skoltech.ru

**Alexander Korotin**
Skolkovo Institute of Science and Technology
Artificial Intelligence Research Institute
Moscow, Russia
a.korotin@skoltech.ru

## Abstract

Aggregating data from multiple sources can be formalized as an *Optimal Transport* (OT) barycenter problem, which seeks to compute the average of probability distributions with respect to OT discrepancies. However, in real-world scenarios, the presence of outliers and noise in the data measures can significantly hinder the performance of traditional statistical methods for estimating OT barycenters. To address this issue, we propose a novel scalable approach for estimating the *robust* continuous barycenter, leveraging the dual formulation of the *(semi-)unbalanced* OT problem. To the best of our knowledge, this paper is the first attempt to develop an algorithm for robust barycenters under the continuous distribution setup. Our method is framed as a min-max optimization problem and is adaptable to *general* cost functions. We rigorously establish the theoretical underpinnings of the proposed method and demonstrate its robustness to outliers and class imbalance through a number of illustrative experiments. Our source code is publicly available at https://github.com/milenagazdieva/U-NOTBarycenters.

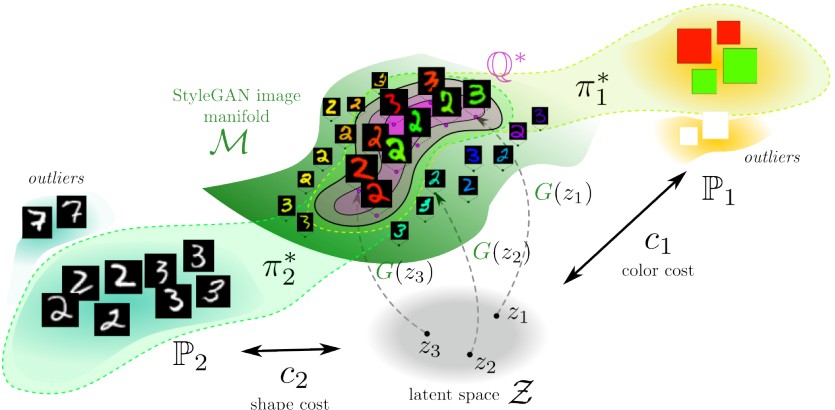

Figure 1: The semi-unbalanced barycenter of distributions of colors ($\mathbb{P}_1$) and digits ($\mathbb{P}_2$) computed by our U-NOTB solver in the latent space of a StyleGAN model pretrained on colored MNIST images of digits '2', '3'. Our solver allows for successful elimination of outliers in the input distributions.

---

[*]Equal contribution

# 1 INTRODUCTION

The world is full of data, and accurate analysis of this data allows for solving a number of problems in the world related to science, medicine, engineering, etc. A common challenge in data analysis arises from two factors: (i) a huge quantity of data, and (ii) the data originating from multiple sources, such as different scientific experiments, hospitals, or engineering trials. One promising way to address these difficulties is to perform *data aggregation*. This means reducing the particular source-specific characteristics and leaving only source-agnostic information for solving a practical case on hand.

Recently, a fruitful branch of research (Li et al., 2020; Fan et al., 2021; Korotin et al., 2022a; Kolesov et al., 2024a;b) has emerged that focuses on addressing the challenge of data aggregation through the framework of the Optimal Transport (OT) barycenter problem. Given a number of reference distributions representing data from different sources, the problem is to discover a geometrically meaningful average of these distributions by minimizing the average OT costs from the reference distributions to the desired one.

Originally introduced for a rather specific setup (quadratic Euclidean OT cost functions) (Agueh & Carlier, 2011), the subfield of OT barycenter has evolved tremendously over the last decade. At present, it is the subject of deep learning and can be adapted to high-dimensional data setups, e.g., images. However, current barycenter research tends to ignore a typical and important property of data appearing in real-world applications: the presence of undesirable noise and outliers (Le et al., 2021). This motivates the development of a *robust* OT barycenter framework which (i) inherits all the advances of current barycenter researches, e.g., permits deep learning and could be adapted to high dims; (ii) wisely deals with aforementioned inconveniences of real-world data. To satisfy both aims, we propose to opt for recent continuous *unbalanced* OT techniques (Gazdieva et al., 2023; Choi et al., 2024b;a) which are naturally aligned with data imperfection.

Our **contributions** are as follows:

1. We propose a novel semi-unbalanced OT (SUOT) barycenter algorithm that is robust to outliers and class imbalance in the reference distributions (§4).

2. We develop a solid theoretical foundation of our proposed approach including the quality bounds for the recovered transport plans (§4.1).

3. We conduct a number of experiments on toy and image data setups to demonstrate the performance of our method and showcase its robustness to outliers and imbalance of classes (§5).

To the best of our knowledge, our work is the first attempt to construct *robust* OT barycenters under *continuous* computational setup (§2.3). Existing works addressing similar problems consider **exclusively** *discrete* settings (Le et al., 2021; Wang et al., 2024).

**Notations.** Throughout the paper, we denote $\overline{K} = \{1, ..., K\}$ for $K \in \mathbb{N}$ and use $s_{[1:K]}$ to denote the tuplet of objects $(s_1, ..., s_K)$. We define $\mathcal{X} \subset \mathbb{R}^{D'}$, $\mathcal{X}_k \subset \mathbb{R}^{D_k}$, $\mathcal{Y} \subset \mathbb{R}^D$ as the compact subsets of Euclidian spaces. We use $\mathcal{C}(\mathcal{Y})$ to denote the set of continuous functions on $\mathcal{Y}$. The sets of absolutely continuous Borel probability (or non-negative) measures on $\mathcal{X}$ are denoted as $\mathcal{P}(\mathcal{X})$ (or $\mathcal{M}_+(\mathcal{X})$). For two measures $\mu_1, \mu_2 \in \mathcal{M}_+(\mathcal{X})$, we write $\mu_1 \ll \mu_2$ to denote that the measure $\mu_1$ is absolutely continuous w.r.t. the measure $\mu_2$. The joint probability distributions on $\mathcal{X} \times \mathcal{Y}$ with the marginals $\mathbb{P}$ and $\mathbb{Q}$ are denoted as $\Pi(\mathbb{P}, \mathbb{Q})$ and usually called the *transport plans*. All joint non-negative measures on $\mathcal{X} \times \mathcal{Y}$ are referred to as $\mathcal{M}_+(\mathcal{X} \times \mathcal{Y})$. For a given measure $\gamma(x, y) \in \mathcal{P}(\mathcal{X} \times \mathcal{Y})$ (or $\mathcal{M}_+(\mathcal{X} \times \mathcal{Y})$), we use $\gamma_x(x), \gamma_y(y)$ to denote its marginals. All probability measures $\gamma(x, y) \in \mathcal{P}(\mathcal{X} \times \mathcal{Y})$ s.t. $\gamma_y = \mathbb{Q}$ for a given probability measure $\mathbb{Q}$ are denoted as $\Pi(\mathbb{Q})$. We use $\gamma(y|x)$ to denote the conditional *probability* measure. For a measurable map $T$, $T_\#$ denotes the corresponding push-forward operator.

The Fenchel conjugate of a function $\psi(u)$ is denoted by $\overline{\psi}(t) \stackrel{\text{def}}{=} \sup_{u \in \mathbb{R}}\{ut - \psi(u)\}$.

# 2 BACKGROUND

In this section, we give an overview of the theoretical concepts related to our paper. In §2.1, we state the SUOT problems and its semi-dual formulation. In §2.2, we formulate the OT/UOT barycenter problem. Then, in §2.3, we describe our computational setup. For details on OT, we refer to (Santambrogio, 2015; Villani et al., 2009; Peyré et al., 2019), unbalanced OT - (Chizat, 2017; Liero et al., 2018; Séjourné et al., 2022), OT barycenters - (Agueh & Carlier, 2011; Chizat, 2023).

To begin with, we introduce the concept of $\psi$-divergences needed for our further derivations.

$\psi$-**divergences for non-negative measures**. Let $\mu_1, \mu_2 \in \mathcal{M}_+(\mathcal{X}')$ be two *non-negative* measures. Then, for a function $\psi : \mathbb{R}_+ \to \mathbb{R}_+ \cup \{\infty\}$, the $\psi$-*divergence* between $\mu_1, \mu_2$ is given by:

$$D_\psi\left(\mu_1\|\mu_2\right) \overset{\text{def}}{=} \int_{\mathcal{X}} \psi\left(\frac{\mu_1(x)}{\mu_2(x)}\right) d\mu_2(x) \text{ if } \mu_1 \ll \mu_2 \text{ and } +\infty \text{ otherwise.}$$

Here, the generator function $\psi(u)$ is assumed to be convex, non-negative, lower semi-continuous, to attain zero uniquely when $u = 1$ and to satisfy the property $\frac{\psi(u)}{u} \overset{u\to\infty}{\to} \infty$. Then, the conjugate $\overline{\psi}(u)$ is also known to be non-negative, and its domain is $\mathbb{R}$, see (Séjourné et al., 2019). Under these assumptions, $D_\psi$ is a valid measure of dissimilarity between two non-negative measures. Well-known examples of $\psi$-divergences include Kullback-Leibler divergence $D_{\text{KL}}$ (Chizat, 2017; Séjourné et al., 2022) and $\chi^2$-divergence $D_{\chi^2}$. For the extended discussion on admissible divergences, we refer to (Gazdieva et al., 2025, Appendix C).

## 2.1 OPTIMAL TRANSPORT

**Classic OT problem.** Consider two probability measures $\mathbb{P}\in\mathcal{P}(\mathcal{X}), \mathbb{Q}\in\mathcal{P}(\mathcal{Y})$ and the cost function $c(x, y) \in \mathcal{C}(\mathcal{X} \times \mathcal{Y})$. Then, the OT problem (Kantorovich, 1942) between $\mathbb{P}$ and $\mathbb{Q}$ is given by

$$\text{OT}_c(\mathbb{P}, \mathbb{Q}) \overset{\text{def}}{=} \inf_{\pi \in \Pi(\mathbb{P}, \mathbb{Q})} \int_{\mathcal{X} \times \mathcal{Y}} c(x, y)\pi(x, y)dxdy. \tag{1}$$

Under the mild assumptions on the distributions $\mathbb{P}$, $\mathbb{Q}$ and cost function $c$, the minimizer $\pi^*$ of (1) always exists but is not guaranteed to be unique, see (Villani et al., 2009). In the special case of the quadratic cost $c(x, y) = \frac{\|x-y\|^2}{2}$, problem (1) reduces to the well-known (squared) Wasserstein-2 distance ($\mathbb{W}_2^2(\mathbb{P}, \mathbb{Q})$). Problem (1) admits a *semi-dual reformulation*:

$$\text{OT}_c(\mathbb{P}, \mathbb{Q}) = \sup_{f \in \mathcal{C}(\mathcal{Y})} \Big[ \int_{\mathcal{X}} f^c(x)d\mathbb{P}(x) + \int_{\mathcal{Y}} f(y)d\mathbb{Q}(y)\Big] \tag{2}$$

where $f^c(x) \overset{\text{def}}{=} \inf_{\mu \in \mathcal{P}(\mathcal{Y})} \int_{\mathcal{Y}} \big(c(x, y) - f(y)\big)d\mu(y)$ is the *c-transform*. Note that the standard $c$-transform in the OT duality is usually defined as $f^c(x) = \inf_{y\in\mathcal{Y}}\{c(x, y) - f(y)\}$. Here, for the future needs, we substitute the classic c-transform by the *weak c-transform* (Backhoff-Veraguas et al., 2019, Theorem 1.3), (Gozlan et al., 2017). This transition is valid since the infimum in weak transform is anyway attained at any $\mu \in \mathcal{P}(\mathcal{Y})$ supported on the $\arg\inf_{y\in\mathcal{Y}}\{c(x, y) - f(y)\}$ set.

**Unbalanced & semi-unbalanced OT problems.** The standard formulation of the OT problem entails various issues (Balaji et al., 2020; Séjourné et al., 2022), such as sensitivity to outliers, inability to deal with class imbalance in the source and target measures and inapplicability to general non-negative measures. Still, we consider the mass transportation between *probability measures* and focus on the first two issues. Fortunately, they can be overcome by relaxing the hard marginal constraints of the OT problem which yields the Unbalanced OT (Chizat, 2017; Liero et al., 2018, UOT) problem. Formally, the UOT problem between the probability measures $\mathbb{P}\in\mathcal{P}(\mathcal{X}), \mathbb{Q}\in\mathcal{P}(\mathcal{Y})$ is

$$\text{UOT}_{c,\psi,\phi}(\mathbb{P}, \mathbb{Q}) = \inf_{\gamma \in \mathcal{M}_+(\mathcal{X} \times \mathcal{Y})} \int_{\mathcal{X} \times \mathcal{Y}} c(x, y)\gamma(x, y)dxdy + D_\psi\left(\gamma_x\|\mathbb{P}\right) + D_\phi\left(\gamma_y\|\mathbb{Q}\right) \tag{3}$$

where $c(x, y) \in \mathcal{C}(\mathcal{X} \times \mathcal{Y})$ is a continuous cost function and $D_\psi, D_\phi$ are $\psi$-divergences over $\mathcal{X}$ and $\mathcal{Y}$, respectively. We can tune the degree of penalizing the marginal distributions mismatch by introducing the *unbalancedness* parameter $\tau > 0$ and considering $D_\psi = \tau D_{\psi'}$, $D_\phi = \tau D_{\phi'}$. Informally, when $\tau \to +\infty$, the corresponding UOT problem tends to classic OT (1). More precisely, the classic OT problem (1) is a special case of the UOT problem (3) where the $\psi$-divergences $D_\psi, D_\phi$ are defined such that $D_{\psi'}\left(\mathbb{P}'\|\mathbb{Q}'\right) = 0$ if $\mathbb{P}' = \mathbb{Q}'$, and $+\infty$ otherwise. Equivalently, it means that the generator functions $\psi, \phi$ are the convex indicators of $\{1\}$ and their conjugates are defined as $\overline{\psi}(t) = \overline{\phi}(t) = t$.

In this paper, we focus on the *semi-unbalanced* OT (SUOT) problem, i.e., the case when only the first marginal constraint in the OT problem (1) is softened:

$$\text{SUOT}_{c,\psi}(\mathbb{P}, \mathbb{Q}) = \inf_{\gamma \in \Pi(\mathbb{Q})} \int_{\mathcal{X} \times \mathcal{Y}} c(x, y)\gamma(x, y)dxdy + D_\psi\left(\gamma_x\|\mathbb{P}\right). \tag{4}$$

Note that the minimizer $\gamma^*$ of (4) always exists thanks to the compactness of the space $\mathcal{X} \times \mathcal{Y}$ (yielding the compactness of $\Pi(\mathbb{Q})$), continuity of the cost function $c(x, y)$ and lower semi-continuity of the function $\psi$ (yielding the lower-semi-continuity of the optimized functional w.r.t. $\gamma$).

The semi-dual SUOT problem is given by

$$\text{SUOT}_{c,\psi}(\mathbb{P}, \mathbb{Q}) = \sup_{f \in \mathcal{C}(\mathcal{Y})} \left\{-\int_{\mathcal{X}} \overline{\psi}(-f^c(x))d\mathbb{P}(x) + \int_{\mathcal{Y}} f(y)d\mathbb{Q}(y)\right\}. \tag{5}$$

Note that under mild assumptions on the cost function $c(x, y)$, optimal potential $f^*$ of (5) always exists, see (Liero et al., 2018, Theorem 4.14). When $\overline{\psi}(t) = t$, problem (5) reduces to classic dual problem (2) for balanced OT problem (1).

## 2.2 (SEMI-)UNBALANCED OPTIMAL TRANSPORT BARYCENTER

Consider probability measures $\mathbb{P}_k \in \mathcal{P}(\mathcal{X}_k)$ and continuous cost functions $c_k(x, y) : \mathcal{X}_k \times \mathcal{Y} \mapsto \mathbb{R}$, $k \in \overline{K}$. Given weights $\lambda_k \geq 0$ s.t. $\sum_{k=1}^{K} \lambda_k = 1$, the *classic* OT barycenter problem consists in finding a minimizer of the sum of OT problems with fixed first marginals $\mathbb{P}_{[1:K]}$:

$$\inf_{\mathbb{Q} \in \mathcal{P}(\mathcal{Y})} \sum_{k=1}^{K} \lambda_k \mathrm{OT}_{c_k}(\mathbb{P}_k, \mathbb{Q}). \tag{6}$$

To ensure the robustness of the barycenter estimation, it is natural to consider the relaxation of the marginals $\mathbb{P}_{[1:K]}$, i.e., substitute the OT problems in (6) with the SUOT problems. Let $\psi_{[1:K]}$ be the set of $\psi$-divergences. Then, the SUOT barycenter problem is

$$\mathcal{L}^* \stackrel{\text{def}}{=} \inf_{\mathbb{Q} \in \mathcal{P}(\mathcal{Y})} \mathcal{B}_u(\mathbb{Q}) \stackrel{\text{def}}{=} \inf_{\mathbb{Q} \in \mathcal{P}(\mathcal{Y})} \sum_{k=1}^{K} \lambda_k \mathrm{SUOT}_{c_k, \psi_k}(\mathbb{P}_k, \mathbb{Q}). \tag{7}$$

Clearly, the SUOT barycenter problem (7) subsumes the conventional OT one (6). We also note that in principle one may substitute OT with UOT problem (3) unbalanced from the both sides (Friesecke et al., 2021; Chung & Phung, 2021; Bonafini et al., 2023). However, this is not practically meaningful. Indeed, the ultimate goal is to get rid of the potential outliers and noises in the input data, not in the barycenter. So there is no need for unbalancedness in the barycenter.

Thanks to (Liero et al., 2018, Corollary 2.9), the functionals $\mathbb{Q} \mapsto \mathrm{SUOT}_{c_k, \psi_k}(\mathbb{P}_k, \mathbb{Q})$ ($k \in \overline{K}$) are convex and lower semi-continuous. Thus, the functional $\mathbb{Q} \mapsto \mathcal{B}_u(\mathbb{Q})$ is convex and lower semi-continuous itself. We note that $\mathcal{P}(\mathcal{Y})$ is weakly compact. Thus, thanks to the Weierstrass theorem (Santambrogio, 2015, Box 1.1), $\mathcal{B}_u(\mathbb{Q})$ admits at least one minimizer meaning that the barycenter $\mathbb{Q}^*$ exists. Still, we can not claim that it is unique since the functional $\mathbb{Q} \mapsto \mathcal{B}_u(\mathbb{Q})$ is not proved to be strictly convex.

## 2.3 COMPUTATIONAL SETUP

In a real-world scenario, the distributions $\mathbb{P}_k$ are not available explicitly and can be assessed only via empirical samples. Assume that we are given $N_k$ empirical samples $x_{[1:N_k]}^k \sim \mathbb{P}_k, k \in \overline{K}$. Our goal is to find approximations $\widehat{\gamma}_k$ of the SUOT plans $\gamma_k^*$ between the measures $\mathbb{P}_k$ and unknown SUOT barycenter $\mathbb{Q}^*$ for the given cost functions and $\psi$-divergences $c_k, \psi_k, \ k \in \overline{K}$. After that, we can use the learned plans to perform the conditional sampling, i.e., derive new points $y \sim \widehat{\gamma}_k(\cdot|x_k)$ from the (approximate) barycenter taking samples $x_k \sim (\widehat{\gamma}_k)_x$ as inputs. Actually, sampling from the left marginals of the learned plans is not an easy task since the marginal can not be easily assessed using the learned plans or potentials. Fortunately, we can deal with this issue using the *rejection sampling* procedure (Forsythe, 1972), see our §4.2 for the additional details.

It is important to note that the learned plans should admit the *new* input samples, i.e., those which are not necessarily present in the training datasets. This setup is typically called *continuous* (Li et al., 2020; Kolesov et al., 2024a; Korotin et al., 2022a) and significantly differs from the discrete one (Peyré et al., 2019; Cuturi & Doucet, 2014). The latter is aimed at solving the barycenter problem between the *empirical* measures which makes its application to new data samples challenging.

## 3 RELATED WORKS

Due to the space constraints, below we give an overview of the most relevant OT barycenter solvers and leave the discussion of the related continuous OT solvers to Appendix B.

**Continuous OT barycenter solvers.** We start with a brief overview of continuous OT barycenter solvers and then proceed to the current state of robust (unbalanced) barycenter research. The continuous OT barycenter methods could be categorized as follows: (Fan et al., 2021; Korotin et al., 2021c) stick to ICNN (Amos et al., 2017) parameterization and work only with quadratic Euclidean costs ($\ell_2^2$); (Korotin et al., 2022a) also deals with $\ell_2^2$ but utilizes fixed point algorithm (Álvarez-Esteban et al., 2016); (Noble et al., 2023) builds upon Schrödinger bridge; (Li et al., 2020; Chi et al., 2023) take the advantage of congruence condition; (Kolesov et al., 2024a;b) empower the congruence condition with Neural OT (Korotin et al., 2023b) and Energy-guided Neural OT (Mokrov et al., 2024) correspondingly. The latter works are the current SOTA in the continuous OT barycenter domain.

The area of robust barycenter computation is much less explored and, to the best of our knowledge, limited exclusively to **discrete** (§2.3) solvers (Chizat et al., 2018; Le et al., 2021; Beier et al., 2023; Séjourné et al., 2023; Wang et al., 2024; Manupriya et al., 2024; Yang & Ding, 2024) based on different principles (MMD regularization/Sinkhorn algorithm/Sliced OT etc.). In contrast, we take a significant step forward and propose the first *continuous* robust barycenter approach with proper theoretical support and practical validation (§4, 5).

## 4 PROPOSED METHOD

In §4.1, we derive and investigate our novel optimization objective for learning SUOT barycenters. In §4.2, we propose the algorithm to solve this problem. The *proofs* for all theoretical results are given in Appendix A.

### 4.1 DERIVING THE OPTIMIZATION OBJECTIVE

To develop a procedure for estimating the barycenter, one may simply substitute the dual form of SUOT problem (5) in the barycenter problem (7) and get a min-max problem. Actually, without additional modifications, it will lead to the min-max-min problem since the $c$-transforms $f_k^{c_k}$ also need to be optimized. In our Theorem 1 below, we present the way for solving the optimization problem (7) in max-min manner and without the optimization over distributions $\mathbb{Q} \in \mathcal{P}(\mathcal{Y})$.

**Theorem 1** (Semi-dual form of SUOT barycenter problem). *The dual form of SUOT barycenter problem* (7) *is given by*

$$\mathcal{L}^* = \sup_{\substack{m \in \mathbb{R}, f_{[1:K]} \in \mathcal{C}^K(\mathcal{Y}) \\ \sum_{k=1}^K \lambda_k f_k \equiv m}} \sum_{k=1}^K \lambda_k \Big[ \int_{\mathcal{X}_k} -\overline{\psi}_k(-f_k^{c_k}(x_k)) d\mathbb{P}_k(x_k) + m \Big]. \tag{8}$$

Hereafter, we say that the potentials $f_{[1:K]}$ are $m$-*congruent* or satisfy the $m$-*congruence condition* if $\sum_{k=1}^K \lambda_k f_k \equiv m$ (for some $m \in \mathbb{R}$). Substituting the definition of the $c$-transform in (8), we get our final optimization objective.

**Corollary 1** (Maximin reformulation for the semi-dual problem (8)). *It holds:*

$$\mathcal{L}^* = \sup_{\substack{m \in \mathbb{R}, \\ f_{[1:K]} \in \mathcal{C}^K(\mathcal{Y}) \\ \sum_{k=1}^K \lambda_k f_k \equiv m}} \inf_{\gamma(\cdot|x_k) \in \mathcal{P}(\mathcal{Y})} \underbrace{\sum_{k=1}^K \lambda_k \left[ \int_{\mathcal{X}_k} -\overline{\psi}_k\left(-\int_{\mathcal{Y}} \big(c_k(x_k, y) - f_k(y)\big) d\gamma_k(y|x_k)\right) d\mathbb{P}_k(x_k) + m \right]}_{\widetilde{\mathcal{L}}(f_{[1:K]}, \{\gamma_k(\cdot|x_k)\}_{[1:K]}, m) \stackrel{def}{=}}. \tag{9}$$

*where the* sup *is taken over* $m \in \mathbb{R}$ *and* $m$-*congruent potentials* $f_{[1:K]} \in \mathcal{C}^K(\mathcal{Y})$, inf $-$ *over conditional plans* $\gamma_k(\cdot|x_k)$.

Interestingly, in the special case when at least one of the $\text{SUOT}_{c_k, \psi_k}$ terms in the right-hand-side of (7) reduces into $\text{OT}_{c_k}$, the $m$-congruence condition in (9) turns into congruence condition $\sum_{k=1}^K \lambda_k f_k \equiv 0$, which typically appears in balanced settings (Kolesov et al., 2024a;b).

**Corollary 2** (Congruence Condition of the Special SUOT barycenter problem). *Suppose that* $\overline{\psi_1}(t) = t$, *i.e.,* $\text{SUOT}_{c_1, \psi_1} = \text{OT}_{c_1}$ *in* (7). *Then, dual form* (9) *turns into the following dual formulation:*

$$\mathcal{L}^* = \sup_{\substack{f_{[1:K]} \in \mathcal{C}^K(\mathcal{Y}) \\ \sum_{k=1}^K \lambda_k f_k \equiv 0}} \inf_{\gamma(\cdot|x_k) \in \mathcal{P}(\mathcal{Y})} \sum_{k=1}^K \lambda_k \left[ \int_{\mathcal{X}_k} -\overline{\psi}_k\left(-\int_{\mathcal{Y}} \big(c_k(x_k, y) - f_k(y)\big) d\gamma_k(y|x_k)\right) d\mathbb{P}_k(x_k) \right]. \tag{10}$$

Theorem 2 below demonstrates the two important properties of the true plans $\gamma^*$ solving the SUOT barycenter problem (7). First, the left marginals of these plans $(\gamma_k^*)_x$ can be characterized via the optimal potentials $f_k^*$ which allows us to find a procedure for sampling from the marginal density during the inference, see §4.2. Second, the Theorem guarantees that the family of optimal plans is indeed one of the solutions to (9).

**Theorem 2** (SUOT barycenter conditional plans are contained in optimal saddle points). *Assume that there exist* $m$ *and* $\{f_k^*(y) \in \mathcal{C}(\mathcal{Y})\}_{k=1}^K$ *which deliver maximum to the problem* (9). *Assume that* $c_k(x, y)$ *are continuous cost functions and* $\overline{\psi}_k$ *are continuously differentiable functions. Let* $\{\gamma_k^*\}_{k=1}^K$ *be a family of optimal SUOT plans between* $\mathbb{P}_k$ *and some barycenter* $\mathbb{Q}^*$. *Then for every plan* $\gamma_k^*$
*(a) the marginal of the plan can be represented as:*

$$d(\gamma_k^*)_x(x) = \nabla \overline{\psi_k}(-(f_k^*)^{c_k}(x)) d\mathbb{P}_k(x); \tag{11}$$

*(b) the corresponding conditional plan* $\gamma_k^*(\cdot|x_k)$ *satisfies*

$$\gamma_k^*(\cdot|x_k) \in \arg\min_{\gamma_k(\cdot|x_k)} \widetilde{\mathcal{L}}(f_{[1:K]}^*, \{\gamma_k(\cdot|x_k)\}_{[1:K]}, m). \tag{12}$$

Theorem (2) shows that for some optimal saddle points $\{(f_k^*, \gamma_k^*)\}_{k=1}^K$ of (9), it holds that $\{\gamma_k^*\}_{k=1}^K$ is the family of true SUOT plans between $\mathbb{P}_{[1:K]}$ and $\mathbb{Q}^*$. Meanwhile, for any family of optimal $f_{[1:K]}^*$, the $\arg\min_{\gamma_k(\cdot|x_k)}$ sets might contain not only optimal SUOT plans $\{\gamma_k^*\}_{k=1}^K$ but other functions as well which is a known *fake solutions* issue of neural OT solvers (Korotin et al., 2023a).

Still, in Theorem 3 below, we show that if the triple $(\widehat{m}, \{\widehat{f}_k\}_{k=1}^K, \{\widehat{\gamma}_k\}_{k=1}^K)$ solves the problem (9) relatively well and some additional assumptions are satisfied, then the recovered conditional plans $\{\widehat{\gamma}(\cdot|x_k)_k\}_{k=1}^K$ are close to those of the true SUOT plans $\{\gamma_k^*(\cdot|x_k)\}_{k=1}^K$. Thanks to this result, we deduce that when Algorithm 1 (§4.2) optimizing (9) converged nearly to the optimum, its solutions are close to the true conditional plans.

**Theorem 3** (Quality bounds for recovered plans). *Consider recovered $\widehat{m}$-congruent potentials $\widehat{f}_{[1:K]}$ s.t. $\widehat{f}_k \in \mathcal{C}(\mathcal{Y}), k \in \overline{K}$ and recovered conditional plans $\{\widehat{\gamma}_k(\cdot|x_k)\}_{k=1}^K, x_k \sim \mathbb{P}_k$ s.t. $\widehat{\gamma}_k(\cdot|x_k) \in \mathcal{P}(\mathcal{Y}), k \in \overline{K}$. Assume that the conditions of Theorem 2 are satisfied, $\mathcal{Y}$ is convex set, and maps $y \mapsto c_k(x_k, y) - \widehat{f}_k(y)$ are $\beta$-strongly convex on $\mathcal{Y}, k \in \overline{K}$. Also, let $0 < \eta \stackrel{def}{=} \nabla \overline{\psi}(-b)$ where $b$ is the upper bound $b = \max\limits_{k \in \overline{K}} \max\limits_{f_k \in \{\widehat{f}_k, f_k^*\}} \max\limits_{x_k \in \mathcal{X}_k, y \in \mathcal{Y}} \{c_k(x_k, y) - f_k(y)\}$. Introduce the duality gaps:*

$$\delta_1(\widehat{f}_{[1:K]}, \widehat{\gamma}_{[1:K]}) \stackrel{def}{=} \widetilde{\mathcal{L}}(\widehat{f}_{[1:K]}, \widehat{\gamma}_{[1:K]}, \widehat{m}) - \inf_{\gamma_k(\cdot|x_k) \in \mathcal{P}(\mathcal{Y})} \widetilde{\mathcal{L}}(\widehat{f}_{[1:K]}, \gamma_{[1:K]}, \widehat{m});$$

$$\delta_2(\widehat{f}_{[1:K]}) \stackrel{def}{=} \mathcal{L}^* - \inf_{\gamma_k(\cdot|x_k) \in \mathcal{P}(\mathcal{Y})} \widetilde{\mathcal{L}}(\widehat{f}_{[1:K]}, \gamma_{[1:K]}, \widehat{m}).$$

*Then, $\frac{2}{\eta\beta}(\delta_1 + \delta_2) \geq \sum_{k=1}^K \lambda_k \int_{\mathcal{X}_k} \mathbb{W}_2^2(\widehat{\gamma}_k(\cdot|x), \gamma_k^*(\cdot|x)) d\mathbb{P}_k(x).$*

Theorem 3 establishes quality bounds for the recovered plans $\{\widehat{\gamma}_k(\cdot|x_k)\}_{k=1}^K$ investigating the *duality gaps*, i.e., the errors for solving inner and outer optimization problems in (9). Note that the strong convexity conditions, in general, are not guaranteed in practice, but are commonly used in the related papers (Fan et al., 2023; Rout et al., 2022), see (Kolesov et al., 2024a, §4) for additional details.

## 4.2 PARAMETRIZATION AND PRACTICAL OPTIMIZATION PROCEDURE

**Parametrization.** To implement the optimization over distributions $\gamma(\cdot|x_k) \in \mathcal{P}(\mathcal{Y})$ $(k \in \overline{K})$ in (9), we consider parametrizing them with stochastic or deterministic functions, using the strategy defined in (Kolesov et al., 2024a, §4.1) and (Korotin et al., 2023a, §4.1).

To realize the optimization over conditional plans $\gamma_k(\cdot|x_k)$ in (9), we define an auxiliary space $\mathcal{S} \subset \mathbb{R}^{D_s}$, atomless distribution $\mathbb{S} \in \mathcal{P}(\mathcal{S})$ and measurable maps $T_k : \mathcal{X}_k \times \mathcal{S} \to \mathcal{Y}$. For every plan $\gamma_k$, we consider the representation using the map $T_k$ s.t. $\gamma_k(\cdot|x) = T_k(x, \cdot)_{\#}\mathbb{S}$. Using the stochastic parametrization of the conditional plans, we can reformulate the optimization objective (9) as:

$$\mathcal{L}^* = \sup_{\substack{m \in \mathbb{R}, \\ f_{[1:K]} \in \mathcal{C}^K(\mathcal{Y}) \\ \sum_{k=1}^K \lambda_k f_k = m}} \inf_{T_{[1:K]}} \underbrace{\sum_{k=1}^K \lambda_k \left[ \int_{\mathcal{X}_k} -\overline{\psi_k} \left( \int_{\mathcal{S}} \Big( f_k(T_k(x_k, s)) - c_k(x_k, T_k(x_k, s)) \Big) d\mathbb{S}(s) \right) d\mathbb{P}_k(x_k) + m \right]}_{\mathcal{L}(f_{[1:K]}, T_{[1:K]}, m) \stackrel{\text{def}}{=}}. \quad (13)$$

We denote the expression under sup inf in (13) as $\mathcal{L}(f_{[1:K]}, T_{[1:K]}, m)$. In some of the setups, the stochasticity of the conditional plans $\gamma(\cdot|x)$ is not needed. Then we can consider a measurable map $T_k : \mathcal{X}_k \to \mathcal{Y}$ which specifies the *deterministic* conditional plans as $\gamma_k(\cdot|x) = \delta_{T_k(x)}(\cdot)$.

For solving (13), we parametrize the maps $T_k$ and potentials $f_k$ $(k \in \overline{K})$ as neural networks $T_{k,\omega} : \mathbb{R}^{D_k} \times \mathbb{R}^{D_s} \mapsto \mathbb{R}^D$ and $f_{k,\theta} : \mathbb{R}^{D_k} \mapsto \mathbb{R}$ with weights $\omega \stackrel{def}{=} \omega_{[1:K]} \in \Omega, \theta \stackrel{def}{=} \theta_{[1:K]} \in \Theta$. Here $\Omega = \Omega_1 \times \Omega_2 \times ... \times \Omega_K$ and $\Theta = \Theta_1 \times \Theta_2 \times ... \times \Theta_K$ are the parameter spaces for the maps and potentials respectively. Note that $\mathbb{R}^{D_s}$ denotes the stochastic dimension which should be omitted in the case of deterministic maps. In order to ensure the $m$-congruence condition for the potentials, we parametrize them as $f_{k,\theta} = g_{k,\theta} - \sum_{n \neq k} \frac{\lambda_n}{\lambda_k(K-1)} g_{n,\theta} + \frac{m}{K\lambda_k}$ where $g_{k,\theta}$ denote the auxiliary neural nets. Here we take inspiration from similar strategies in (Li et al., 2020; Kolesov et al., 2024b;a).

**Training.** Recall that we work in the continuous setup of the barycenter problem, i.e., the distributions $\mathbb{P}_{[1:K]}$ are accessible only through the empirical samples of data. Thus, we opt to estimate the objective (13) from samples using the Monte-Carlo method. Specifically, the objective is optimized using stochastic gradient descent-ascent algorithm over random batches of samples from the distributions $\mathbb{P}_k$ and auxiliary distribution $\mathbb{S}$[1]. For simplicity, we replace the minimization of the objective $\mathcal{L}(f_{\theta,[1:K]}, T_{\omega,[1:K]})$ w.r.t. $T_{\omega,[1:K]}$ with the direct minimization of the $c$-transform $\int_{\mathcal{X}_k} \int_{\mathcal{S}} \Big( c_k(x_k, T_k(x_k, s)) - f_k(T_k(x_k, s)) \Big) d\mathbb{S}(s) d\mathbb{P}_k(x_k)$ which has the same set of minimizers. We detail our optimization procedure in Algorithm 1.

**Inference.** Our approach for SUOT barycenter estimation relaxes the marginal constraints related to

---

[1]Sampling from the stochastic distribution is omitted when we deal with the deterministic maps $T_{\omega,[1:K]}$.

---

**Algorithm 1:** SUOT Barycenter with Semi-Unbalanced Neural Optimal Transport

---

**Input:** Distributions $\mathbb{P}_{1:K}, \mathbb{S}$ accessible by samples; transport costs $c_k : \mathcal{X}_k \times \mathcal{Y} \mapsto \mathbb{R}$; maps $T_{k,\omega}$ and $m$-congruent potentials $f_{k,\theta}$, $k \in \overline{K}$; number $N_T$ of inner iterations; batch sizes.
**Output:** Trained (stochastic) maps $T_{[1:K],\omega^*}$ approximating SUOT plans between $\mathbb{P}_k$ and barycenter $\mathbb{Q}^*$.
**repeat**

> Sample batches $X_k \sim \mathbb{P}_k$, $k \in \overline{K}$; For each $x_k \in X_k$ sample an auxiliary batch $S[x_k] \sim \mathbb{S}$;
>
> $\widehat{\mathcal{L}_f} \leftarrow - \sum_{k \in \overline{K}} \lambda_k \left[ \frac{1}{|X_k|} \sum_{x_k \in X_k} \overline{\psi}\left( \sum_{s_k \in S[x_k]} \frac{f(T_k(x_k,s)) - c(x_k, T_k(x_k,s))}{|S[x_k]|} \right) + m \right]$;
>
> Update $\theta$ by using $\frac{\partial \widehat{\mathcal{L}_f}}{\partial \theta}$ to *maximize* $\widehat{\mathcal{L}_f}$;
>
> **for** $n_T = 1, 2, \ldots, N_T$ **do**
>
> > Sample batches $X_k \sim \mathbb{P}_k$; For each $x_k \in X_k$:
> > > sample an auxiliary batch $S[x_k] \sim \mathbb{S}$;
> >
> > $\widehat{\mathcal{L}_{T_k}} \leftarrow \frac{1}{|X_k|} \sum_{x_k \in X_k} \sum_{s_k \in S[x_k]} \frac{c(x_k, T_k(x_k,s)) - f(T_k(x_k,s))}{|S[x_k]|}$
> >
> > Update $\omega$ by using $\frac{\partial \widehat{\mathcal{L}_T}}{\partial \omega}$ to *minimize* $\widehat{\mathcal{L}_T}$;

**until** *not converged*;

---

the input distributions $\mathbb{P}_k$. It means that during the inference, the input points $x_k$ should be sampled not from the distributions $\mathbb{P}_k$ but from the left marginals of the learned plans $(\widehat{\gamma}_k)_x$. However, sampling from these marginals is not available explicitly and requires the usage of the established techniques for sampling from the *reweighted* distributions, e.g., the *rejection sampling* technique. We apply this technique in the following manner.

Thanks to our Theorem 2 (a), we are able to approximate the fraction of the densities of measures $(\gamma_k^*)_x$ and $\mathbb{P}_k$: $\frac{d(\gamma_k^*)_x(x)}{d\mathbb{P}_k(x)} \approx \nabla\overline{\psi}(-(\widehat{f}_k)^c(x))$ where $\widehat{f}_k$ denotes the potential learned during training. Thus, we can (1) generate new samples $X_k \sim \mathbb{P}_k$ and samples of the same shape from the uniform distribution $u_k \sim U$; (2) calculate the constant $c \overset{\text{def}}{=} \max \left( \nabla\overline{\psi}(-(\widehat{f}_k)^c(x)) \right)$; (3) accept the samples $X_k'$ which satisfy the inequality $u_k \leq \nabla\overline{\psi}(-(\widehat{f}_k)^c(X_k'))$ and reject the others.

## 5 EXPERIMENTS

In this section, we evaluate our model through various experiments. In §5.1, we compare the numerical accuracy of our method against baseline methods. In §5.2, we investigate two key properties of our **U-NOTB** method: robustness to class imbalance and robustness to outliers. Specifically, we show that these properties can be controlled by adjusting the unbalanced parameter $\tau$. In §5.3, we evaluate our model for the general costs, leveraging shape and color-invariant cost on the image dataset, other than the quadratic cost. We provide the *details about settings and baselines* in Appendix C. In Appendices D.1, D.2, we test our solver in the balanced/semi-unbalanced OT barycenter problem for Gaussian distributions with computable *ground-truth solutions*. Additionally, in Appendix D.3, we illustrate another *interesting application* of U-NOTB in higher dimensions showing its ability to manipulate images through interpolating its distributions on the image manifolds. The code for our model is written using PyTorch framework and is publicly available at

> https://github.com/milenagazdieva/U-NOTBarycenters.

### 5.1 BARYCENTER EVALUATION ON SYNTHETIC DATASETS

**Baselines.** We aim to evaluate whether our model accurately estimates the optimal barycenter $\mathbb{Q}^*$ and the corresponding transport maps $T^*_{[1:K]}$. Because our work is the first attempt to address the continuous SUOT barycenters, there are no existing baselines for direct comparison. Therefore, we carefully designed *two baseline models for comparison with our approach*. These baseline models are motivated by the equivalence between the OT barycenter of two distributions and the interpolation between them (Villani et al., 2009). Each baseline model is derived from two approaches for learning the unbalanced transport map $T_{uot}$ between $\mathbb{P}_1$ and $\mathbb{P}_2$: (i) the semi-dual UOT model (UOTM) (Choi et al., 2024a) and (ii) the OT Map model (OTM) (Rout et al., 2022) combined with the mini-batch UOT sampling (Eyring et al., 2024). Because the SUOT problem is equivalent to the OT problem

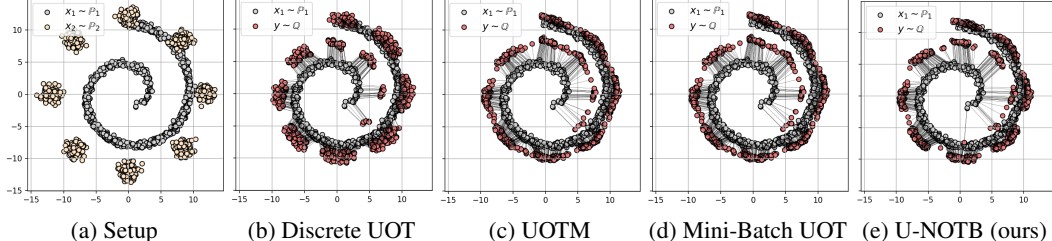

| (a) Setup | (b) Discrete UOT | (c) UOTM | (d) Mini-Batch UOT | (e) U-NOTB (ours) |

Figure 2: Transport map $T_1(x_1)$, $x_1 \sim \mathbb{P}_1$ and UOT barycenter distribution $\mathbb{Q} = T_{1\#}\mathbb{P}_1$ obtained by Discrete UOT, UOTM, Mini-batch UOT, and our method in *Spiral → Gaussian Mixture*.

| Data | Metric | UOTM | Mini-Batch UOT | U-NOTB (ours) |
|---|---|---|---|---|
| *Sprial → Gaussian Mixture* | $\mathcal{L}_2$ | 0.65 | 0.31 | **0.25** |
| | $\mathbb{W}_2$ | 0.29 | 0.15 | **0.12** |
| *Moon → Spiral* | $\mathcal{L}_2$ | 0.89 | 0.64 | **0.52** |
| | $\mathbb{W}_2$ | **0.59** | 0.93 | 0.81 |

Table 1: Comparison on Benchmarks on Toy datasets.

between the rescaled distributions, we conduct the displacement interpolation using this $T_{uot}$ to approximate the barycenter. Specifically, we consider the following SUOT barycenter problem:

$$\inf_{\mathbb{Q} \in \mathcal{P}(\mathcal{Y})} \{\lambda_1 \text{OT}_{c_1}(\mathbb{P}_1, \mathbb{Q}) + \lambda_2 \text{SUOT}_{c_2}(\mathbb{P}_2, \mathbb{Q})\}, \quad (14)$$

where $c_1, c_2$ are the quadratic cost functions $c_k(x, y) = \frac{\|x-y\|^2}{2}$, $\lambda_1 = \lambda_2 = \frac{1}{2}$. Let $T_1 \overset{\text{def}}{=} \lambda_1 Id + \lambda_2 T_{uot}$. Then, $T_{1\#}\mathbb{P}_1$ and $T_1$ approximate the barycenter distribution $\mathbb{Q}^*$ and the transport map from $\mathbb{P}_1$ to $\mathbb{Q}^*$, respectively. Note that we should set the $\text{OT}_{c_1}$ distance for the first distribution $\mathbb{P}_1$ to ensure that the estimated barycenter $\mathbb{Q} = T_{1\#}\mathbb{P}_1$ has a unit mass in two baselines. Since the OT problem is a specific case of the SUOT one, our method can be naturally applied to this case. We test U-NOTB on the synthetic dataset of *Spiral-to-Gaussian Mixture* (S→GM) (Fig. 2a).

**Experimental Results.** Fig. 2 presents a qualitative visualization of the results from our model and two baseline methods. Since there is no closed-form solution for the continuous SUOT barycenter problems, we consider the SUOT barycenter between two discrete empirical distributions, obtained from the POT (Flamary et al., 2021) library, as the ground-truth solution $\mathbb{Q}^*$ and $T_1^*$ (Fig. 2b). In Fig. 2c-2e, we visualize the spiral distribution $\mathbb{P}_1$, the estimated transport map $T_1$ and barycenter $\mathbb{Q} = T_{1\#}\mathbb{P}_1$. Note that, because we set $\text{OT}_{c_1}$ for $\mathbb{P}_1$ in Eq. (14), the rejection sampling is not applied in this case. Fig. 2 shows that our model and other baseline methods present decent results.

Table 1 provides a quantitative evaluation of our model for a more detailed assessment. The approximate barycenter $\mathbb{Q}_{\omega^*}$ and the transport map $T_{1,\omega^*}$ from each method are evaluated from two perspectives. First, the distribution error between $\mathbb{Q}_{\omega^*}$ and $\mathbb{Q}^*$ is evaluated by measuring the Wasserstein-2 distance $\mathbb{W}_2^2(\mathbb{Q}_{\omega^*}, \mathbb{Q}^*)$ ($\mathbb{W}_2$ *metric*). Second, the optimality error of the transport map is assessed by measuring $\mathcal{L}_2$-norm between the transport maps ($\mathcal{L}_2$ *metric*), i.e., $\int_{\mathcal{X}_1} \|T_{1,\omega^*}(x_1) - T_1^*(x_1)\|_2^2 d\mathbb{P}_1(x_1)$. As shown in Table 1, our model achieves the most accurate barycenter estimation in three out of four scores, evaluated using two metrics ($\mathbb{W}_2$ and $\mathcal{L}_2$) across two synthetic datasets.

## 5.2 ROBUST BARYCENTER ESTIMATION UNDER OUTLIER AND CLASS IMBALANCE

In this section, we examine two key properties of our U-NOTB model: (i) robustness to class imbalancedness, (ii) robustness to outliers. The value of OT functional $\text{OT}_c(\cdot, \cdot)$ is known to be sensitive to outliers and class imbalance problems. Hence, the OT barycenter is also largely affected by these factors. To address this sensitivity, the UOT problem extends the traditional OT by relaxing the constraint on marginal densities. This relaxation introduces two notable characteristics. First, even in the presence of class imbalances, the reweighting process allows the model to appropriately align with the relevant modes in a reasonable manner (Eyring et al., 2024). Second, the model exhibits robustness to outliers, maintaining reliable transport despite their presence (Balaji et al., 2020; Choi et al., 2024a). The goal of this section is twofold: (1) to investigate whether these two properties are also observed in our U-NOTB model by comparing it with its OT counterpart (Kolesov et al., 2024a) and (2) to demonstrate that this robustness can be controlled through the unbalancedness parameter $\tau$.

**Robustness to Class Imbalance.** The class imbalance issue refers to the case where each class has a different proportion across $\mathbb{P}_k$. This class imbalance problem can lead to undesirable behaviour in the standard OT barycenter. For example, in Fig. 3, we consider the upper-modes and bottom-modes for each distribution as class 1 and 2, respectively. In $\mathbb{P}_1$, 75% of the data is concentrated in class 1

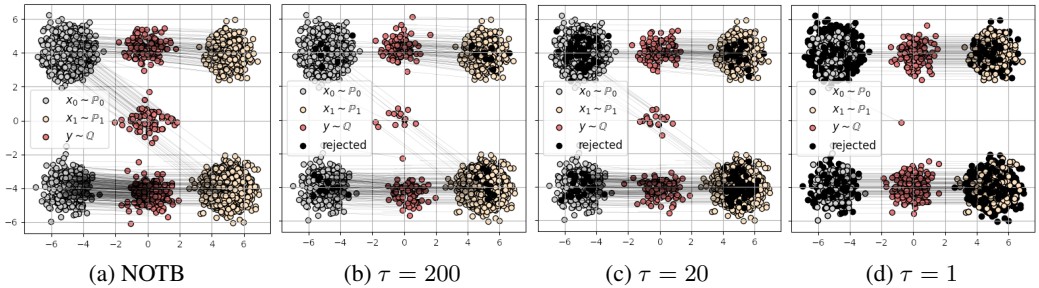

|  |  |  |  |
|---|---|---|---|
| (a) NOTB | (b) $\tau = 200$ | (c) $\tau = 20$ | (d) $\tau = 1$ |

Figure 3: Conditional plans $\gamma_k(y|x)$ and barycenter $\mathbb{Q}$ obtained by NOTB and our method in *Gaussian Mixture* barycenter experiment. We evaluate on various unbalancedness parameter $\tau \in \{1, 20, 200\}$.

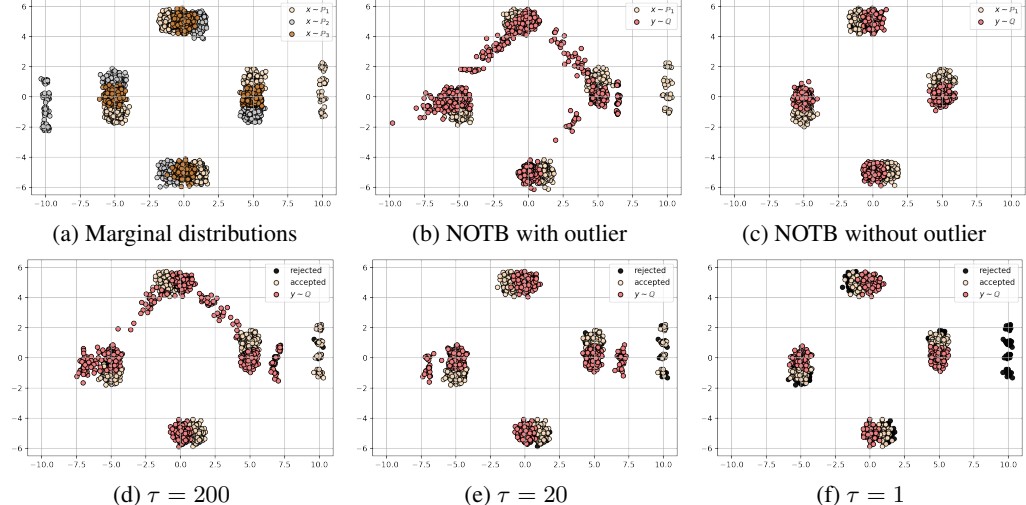

|  |  |  |
|---|---|---|
| (a) Marginal distributions | (b) NOTB with outlier | (c) NOTB without outlier |
| (d) $\tau = 200$ | (e) $\tau = 20$ | (f) $\tau = 1$ |

Figure 4: Learned barycenter $\mathbb{Q}$ obtained from $x \sim \mathbb{P}_1$ by NOTB and our method on *Gaussian Mixture* with *5% outliers*. The evaluation is conducted for various unbalancedness parameter $\tau \in \{1, 20, 200\}$. For comparison, we additionally trained NOTB without outliers, as shown in subfigure (c).

(upper-left white dots) while only 25% is concentrated in class 1 for $\mathbb{P}_1$ (upper-right yellow dots). Therefore, the balanced OT barycenter (Kolesov et al., 2024a, NOTB) generates a mode between two majority modes of $\mathbb{P}_k$, which can be an undesirable phenomenon for the barycenter (Fig. 3a).

As shown in Fig. 3, we evaluated our model for diverse unbalancedness parameter $\tau \in \{1, 20, 100\}$ (Eq. 3). Additionally, for comparison, we conducted experiments under a balanced setting using the OT counterpart. It is important to note that as $\tau$ increases, the UOT problem converges to the OT problem, allowing for less flexibility in marginal distributions (Choi et al., 2024b). When $\tau = 1$, our unbalanced barycenter demonstrates robust results by generating barycenter modes between the pair of the closest modes of the $\mathbb{P}_1$ and $\mathbb{P}_2$. Our model rejects contour samples from majority modes, reweighting them to better align the corresponding modes. Moreover, our unbalanced barycenter offers controllability of this robustness through $\tau$. As $\tau$ increases, the flexibility on marginal distributions decreases in the UOT problem. Thus, when $\tau = 200$, our model accepts nearly all samples, making the UOT barycenter closely resemble the OT barycenter. We believe this adaptability of the UOT barycenter offers practical benefits in real-world scenarios.

**Robustness to Outliers.** To evaluate the robustness to outliers, we conducted experiments on a synthetic dataset that included a small proportion of outliers. As depicted in Fig. 4, the dataset comprises three marginal distributions: $\mathbb{P}_1$ (beige dots), $\mathbb{P}_2$ (gray dots), and $\mathbb{P}_3$ (orange dots). Outliers, constituting 5% of the marginal $\mathbb{P}_1$ and $\mathbb{P}_2$, were added to the outermost point of each marginal. We evaluated our model for various unbalancedness parameters $\tau$ and compared it with the balanced OT counterpart. Similar to the robustness to class imbalance experiments, we expect that smaller values of $\tau$ will provide more robustness to outliers by allowing higher flexibility in marginal distributions.

For the comparative analysis, we also tested the NOTB approach for OT barycenter estimation under the data without outliers. This experiment aims to demonstrate the desired behaviour of barycenter when there are no outliers in the dataset. In Fig. 4, we see that for $\tau = 1$, our U-NOTB closely aligns with the OT barycenter in this outlier-free scenario. This result shows that our model offers robustness

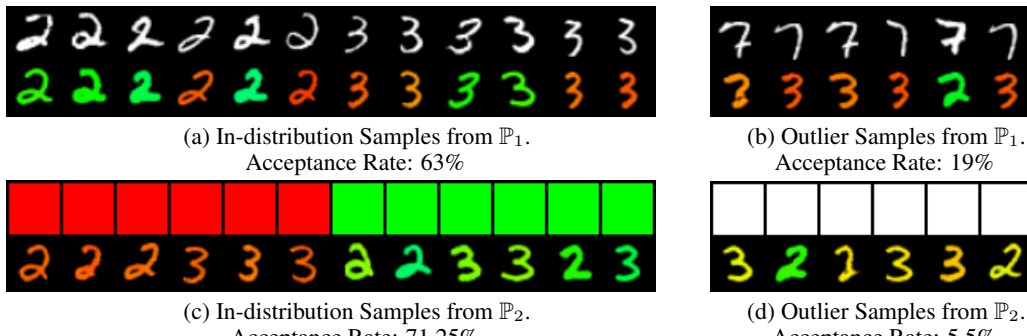

(a) In-distribution Samples from $\mathbb{P}_1$.
Acceptance Rate: 63%

(b) Outlier Samples from $\mathbb{P}_1$.
Acceptance Rate: 19%

(c) In-distribution Samples from $\mathbb{P}_2$.
Acceptance Rate: 71.25%

(d) Outlier Samples from $\mathbb{P}_2$.
Acceptance Rate: 5.5%

Figure 5: Examples of $x_1 \sim \mathbb{P}_1$ (grayscale digits) and $x_2 \sim \mathbb{P}_2$ (color images) and its corresponding barycenter $y \sim \mathbb{Q}$ samples (colored-digits) in *shape-color* experiment.

to outliers by rejecting outlier samples at smaller $\tau$. As $\tau$ increases, our model accepts a higher proportion of outliers. Hence, the outliers on the left and right begin to influence the barycenter, making the modes of barycenter lie between the majority modes of distributions $\mathbb{P}_k$. In summary, reducing $\tau$ enhances outlier robustness, while increasing $\tau$ yields a more precise barycenter by incorporating all data points. We believe that the flexibility to adjust $\tau$ offers promising potential for broader applications, where different levels of tolerance to outliers may be required.

### 5.3 SHAPE-COLOR EXPERIMENT

In this section, we illustrate one interesting example demonstrating how the UOT barycenter problem can be applied using a general cost. We designed the problem at general costs when there are some undesirable outliers in the marginal distributions, see Fig. 1 for visualization of the setup.

- **Shape distribution $\mathbb{P}_1$.** The first marginal distribution consists of grayscaled images of digits '2' (49% of training dataset), '3' (50%) and '7' (1% - *outliers*) of MNIST data.

- **Color distribution $\mathbb{P}_2$.** The second marginal distribution consists of three color points: red (probability mass $p_0 = 0.495$), green ($p_1 = 0.495$), white ($p_2 = 0.01$ - *outliers*).

- **Manifold.** We pretrained the StyleGAN (Karras et al., 2019) generator $G : \mathcal{Z} \to \mathbb{R}^{3 \times 32 \times 32}$ on the colored MNIST dataset of digits '2' and '3'. Let $E_\theta^k$ be the network which encodes each marginal $\mathbb{P}_k$ to the latent $\mathcal{Z}$. Throughout the experiment, we transport $x_k \sim \mathbb{P}_k$ to the barycenter point $y_k$ by $y_k = G \circ E_\theta^k(x_k)$, thus confining the transformed images to the colored MNIST of digits '2' and '3'.

- **General Transport costs.** For the first marginal sample $x_1 \sim \mathbb{P}_1$ and its corresponding barycenter point $y_1$, we use the following *shape-preserving cost*: $c_1(x_1, y_1) = \frac{1}{2}\|x_1 - H_g(y_1)\|^2$, where $H_g$ is a decolorization operator. Moreover, for the second marginal sample $x_2 \sim \mathbb{P}_2$ and the corresponding barycenter point $y_2$, we use the following *color-preserving cost*: $c_2(x_2, y_2) = \frac{1}{2}\|x_2 - H_c(y_2)\|^2$, where $H_c$ is a color projection operator defined in (Kolesov et al., 2024a).

Our results for unbalancedness $\tau = 10$ are presented in Figure 5. We demonstrate the examples of learned mapping $(x_k, T_k(x_k, s))$ from in-distribution points $x_k$, see Fig. 5a, 5c, and from outliers, see Fig. 5b, 5d. Expectedly, the learned in-distribution mappings preserve the shape ($\mathbb{P}_1 \to \mathbb{Q}^*$) and color ($\mathbb{P}_2 \to \mathbb{Q}^*$). In turn, the outlier mappings have no reasonable interpretation.

Along with the qualitative performance, we report the *acceptance rates* of input points $x_k$, see the explanation of our inference procedure in §4.2. Importantly, the acceptance of outliers is much smaller compared to in-distribution samples. In particular, if $x_1$ is a grayscaled digit '7', it will be accepted (and processed through the mapping $T_1$) only with probability 0.19, while for grayscaled digits '2' and '3' the same figure reads as 0.63. Thus, the experiment showcases the applicability of our robust barycenter methodology for non-trivial cases with non-Euclidean OT costs.

## 6 DISCUSSION

Our work continues the recent and fruitful branch of OT barycenter research. We present the *first* attempt to build *robust* OT barycenters under continuous setup based on unbalanced OT formulation. From now on, the barycenter researchers have a new deep learning tool at their disposal that gives them the ability to deal with imperfect data (with outliers/noise/class imbalance) at a large scale. We believe that our proposed method will further expand the toolbox of deep learning practitioners. We discuss the limitations of our method and future research directions in Appendix E.

**Reproducibility.** For transparency and reproducibility, we make our source code publicly available at `https://github.com/milenagazdieva/U-NOTBarycenters`. The code contains a `README` file with further instructions to reproduce our experiments.

**Code of Ethics.** This paper presents work whose goal is to advance the field of Machine Learning. There are many potential societal consequences of our work, none of which we feel must be specifically highlighted here.

ACKNOWLEDGEMENTS. The work of Skoltech was supported by the Analytical center under the RF Government (subsidy agreement 000000D730321P5Q0002, Grant No. 70-2021-00145 02.11.2021). Jaemoo Choi acknowledges the support by National Research Foundation of Korea (NRF) grant funded by the Korea government (MSIT) [RS-2024-00410661].

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

# A PROOFS

## A.1 PROOFS OF THEOREM 1 AND COROLLARY 1

*Proof of Theorem 1.* Substituting the dual form (5) into (13), we can formulate the barycenter problem as finding

$$\mathcal{L}^* = \min_{\mathbb{Q} \in \mathcal{P}(\mathcal{Y})} \underbrace{\sup_{f_1,\ldots,f_K \in \mathcal{C}(\mathcal{Y})} \sum_{k=1}^{K} \lambda_k \Big[ \int_{\mathcal{X}_k} -\overline{\psi}_k(-f_k^c(x)) d\mathbb{P}_k(x_k) + \int_{\mathcal{Y}} f_k(y) d\mathbb{Q}(y) \Big]}_{\mathcal{F}(\mathbb{Q}, f_{[1:K]}) \overset{\text{def}}{=}} \quad (15)$$

where we replace inf with min thanks to the existence of SUOT barycenter, see §2.2.

Note that the compactness of the space $\mathcal{Y}$ yields the weak compactness of $\mathcal{P}(\mathcal{Y})$. At the same time, the functional $\mathbb{Q} \mapsto \mathcal{F}(\mathbb{Q}, f_{[1:K]})$ is continuous and linear in $\mathbb{Q}$. At the same time, it is concave in $f_{[1:K]}$. Indeed, the convex conjugate of the generator function $\psi$ is convex and non-decreasing by definition, the $c$-transform operation is convex, see (Kolesov et al., 2024b, Proposition A.1 (iii)). Thus, the superposition of a concave, non-increasing function $-\overline{\psi}_k(\cdot)$ and concave function $-f_k^c(\cdot)$, i.e., $-\overline{\psi}_k(-f_k^c(\cdot))$ is a concave function as well (for every $k \in \overline{K}$). Thus, we can apply the minimax theorem (Terkelsen, 1972, Corollary 1) and swap min and sup in (15):

$$\mathcal{L}^* = \sup_{f_1,\ldots,f_K \in \mathcal{C}(\mathcal{Y})} \min_{\mathbb{Q} \in \mathcal{P}(\mathcal{Y})} \sum_{k=1}^{K} \lambda_k \Big[ \int_{\mathcal{X}_k} -\overline{\psi}_k(-f_k^c(x_k)) d\mathbb{P}_k(x_k) + \int_{\mathcal{Y}} f_k(y) d\mathbb{Q}(y) \Big] =$$

$$\sup_{f_1,\ldots,f_K \in \mathcal{C}(\mathcal{Y})} \Big[ \sum_{k=1}^{K} \lambda_k \int_{\mathcal{X}_k} -\overline{\psi}_k(-f_k^c(x_k)) d\mathbb{P}_k(x_k) + \min_{\mathbb{Q} \in \mathcal{P}(\mathcal{Y})} \int_{\mathcal{Y}} \underbrace{\sum_{k=1}^{K} \lambda_k f_k(y)}_{\overline{f}(y) \overset{\text{def}}{=}} d\mathbb{Q}(y) \Big] =$$

$$\sup_{f_1,\ldots,f_K \in \mathcal{C}(\mathcal{Y})} \underbrace{\Big[ \sum_{k=1}^{K} \lambda_k \int_{\mathcal{X}_k} -\overline{\psi}_k(-f_k^c(x_k)) d\mathbb{P}_k(x_k) + \min_{\mathbb{Q} \in \mathcal{P}(\mathcal{Y})} \int_{\mathcal{Y}} \overline{f}(y) d\mathbb{Q}(y) \Big]}_{\mathcal{G}(f_1,\ldots f_K) \overset{\text{def}}{=}}. \quad (16)$$

Now we use the following fact: $\inf_{\mathbb{Q} \in \mathcal{P}(\mathcal{Y})} \int_{\mathcal{Y}} \overline{f}(y) d\mathbb{Q}(y) = \inf_{y \in \mathcal{Y}} \overline{f}(y)$. Assume that the true value $m \overset{\text{def}}{=} \min_{y \in \mathcal{Y}} \overline{f}(y)$ is known. Then we can restrict sup in (16) to the potentials $f_{[1:K]}$ satisfying the *m-congruence* condition: $\sum_{k=1}^{K} \lambda_k f_k \equiv m$. Indeed, for each tuple $(f_1, \ldots, f_K)$, let us consider the tuple $(\widetilde{f}_1, \ldots, \widetilde{f}_K) \overset{\text{def}}{=} (f_1, \ldots, f_K + \frac{m - \overline{f}}{\lambda_K})$ satisfying the congruence condition. For this tuple, $\inf_{y \in \mathcal{Y}} \sum_{k=1}^{K} \lambda_k \widetilde{f}_k = m$. Besides, we get:

$$\mathcal{G}(\widetilde{f}_1, \ldots, \widetilde{f}_K) - \mathcal{G}(f_1, \ldots f_K) = \lambda_K \int_{\mathcal{X}_K} [-\overline{\psi}_K(-\widetilde{f}_K^c(x_K)) + \overline{\psi}_K(-f_K^c(x_K))] d\mathbb{P}_K(x_K) + \cancel{m} - \cancel{m} =$$

$$\lambda_K \int_{\mathcal{X}_K} \Big[ \overline{\psi}_K(-f_K^c(x_K)) - \overline{\psi}_K(-(f_K + \frac{m - \overline{f}}{\lambda_K})^c(x_K)) \Big] d\mathbb{P}_K(x_K) \geq 0. \quad (17)$$

Here the inequality in line (17) follows from the properties of the $c$-transform function and convex conjugate of the divergence' generator function $\overline{\psi}$. First, $c$-transform is a decreasing function of its argument, see (Kolesov et al., 2024b, Proposition A.1 (i)). Thus,

$$f_K + \frac{m - \overline{f}}{\lambda_K} \leq f_K + \cancel{\frac{m - m}{\lambda_K}} = f_K \implies (f_K + \frac{m - \overline{f}}{\lambda_K})^{c_K} \geq f_K^{c_K}.$$

Second, the convex conjugate $\overline{\psi}_K$ is a non-decreasing function which yields

$$-f_K^{c_K} \geq -(f_K + \frac{m - \overline{f}}{\lambda_K})^{c_K} \implies \overline{\psi}_K(-f_K^{c_K}) \geq \overline{\psi}_K(-(f_K + \frac{m - \overline{f}}{\lambda_K})^{c_K}).$$

From this, the inequality (17) becomes obvious. It means that the transition to congruent tuple is fair for the known value $m$. It is important to note that in practice this value is not given and should be optimized. Thus, the final optimization objective can be formalized as

$$\mathcal{L}^* = \sup_{\substack{m \in \mathbb{R}, f_{[1:K]} \in \mathcal{C}^K(\mathcal{Y}) \\ \sum_{k=1}^K \lambda_k f_k \equiv m}} \mathcal{G}(f_1, ..., f_K) = \sup_{\substack{m \in \mathbb{R}, f_{[1:K]} \in \mathcal{C}^K(\mathcal{Y}) \\ \sum_{k=1}^K \lambda_k f_k \equiv m}} \left[ \sum_{k=1}^K \lambda_k \int_{\mathcal{X}_k} -\overline{\psi}_k(-f_k^c(x_k)) d\mathbb{P}_k(x_k) + m \right] \quad (18)$$

where (18) coincides with (9). $\qquad\square$

*Proof of Corollary 1.* By substituting the definition of the $c$-transform in (8), we get

$$\mathcal{L}^* = \sup_{\substack{m \in \mathbb{R}, f_{[1:K]} \in \mathcal{C}^K(\mathcal{Y}) \\ \sum_{k=1}^K \lambda_k f_k \equiv m}} \sum_{k=1}^K \lambda_k \left[ \int_{\mathcal{X}_k} -\overline{\psi}_k \left( -\inf_{\mu(y) \in \mathcal{P}(\mathcal{Y})} \int_{\mathcal{Y}} (c_k(x_k, y) - f_k(y)) d\mu(y) \right) d\mathbb{P}_k(x_k) + m \right] = \quad (19)$$

$$\sup_{\substack{m \in \mathbb{R}, f_{[1:K]} \in \mathcal{C}^K(\mathcal{Y}) \\ \sum_{k=1}^K \lambda_k f_k \equiv m}} \sum_{k=1}^K \lambda_k \left[ \int_{\mathcal{X}_k} \inf_{\mu(y) \in \mathcal{P}(\mathcal{Y})} \left\{ -\overline{\psi}_k \left( -\int_{\mathcal{Y}} (c_k(x_k, y) - f_k(y)) d\mu(y) \right) \right\} d\mathbb{P}_k(x_k) + m \right]. \quad (20)$$

In transition from (19) to (20), we interchange the operation of taking inf over $\mu(y) \in \mathcal{P}(\mathcal{Y})$ and $\overline{\psi}_k$. This is possible since $-\overline{\psi}_k$ is monotone non-increasing and infimum in (19) is attained.

Now we note that the map

$$(x_k, \mu) \mapsto \left\{ -\overline{\psi}_k \left( -\int_{\mathcal{Y}} (c_k(x_k, y) - f_k(y)) d\mu(y) \right) \right\} \quad (21)$$

is measurable and bounded from below. Indeed, recall that the spaces $\mathcal{X}_k$, $\mathcal{Y}$ and $\mathcal{P}(\mathcal{Y})$ are compact, the potentials $f_k(y)$ and cost functions $c_k(x_k, y)$ are continuous. Therefore, the function $(x_k, \mu) \mapsto -\int_{\mathcal{Y}} (c_k(x_k, y) - f_k(y)) d\mu(y)$ is bounded from above, i.e. $\exists B_k \in \mathbb{R}$:

$$\forall (x_k, \mu) \in \mathcal{X}_k \times \mathcal{P}(\mathcal{Y}) : -\int_{\mathcal{Y}} (c_k(x_k, y) - f_k(y)) d\mu(y) \le B_k.$$

Since $-\overline{\psi}_k$ is monotone non-increasing and $\text{dom}(\overline{\psi}_k) = (-\infty, \underbrace{\lim_{u \to \infty} \frac{\psi(u)}{u}}_{=+\infty}) = \mathbb{R}$, then:

$$\forall (x_k, \mu) \in \mathcal{X}_k \times \mathcal{P}(\mathcal{Y}) : -\overline{\psi}_k \left( -\int_{\mathcal{Y}} (c_k(x_k, y) - f_k(y)) d\mu(y) \right) \ge -\overline{\psi}_k(B_k) > -\infty.$$

The latter justifies that (21) is indeed bounded from below; its measurability is obvious. Thanks to the properties of this map, we can use the rule of interchange between integral and inf (Bertsekas & Shreve, 1996, Propositions 7.27 and 7.50) and get

$$(20) =$$

$$\sup_{\substack{m \in \mathbb{R}, f_{[1:K]} \in \mathcal{C}^K(\mathcal{Y}) \\ \sum_{k=1}^K \lambda_k f_k \equiv m}} \sum_{k=1}^K \lambda_k \left[ \inf_{\gamma_k(\cdot|x_k) : \mathcal{X}_k \mapsto \mathcal{P}(\mathcal{Y})} \int_{\mathcal{X}_k} -\overline{\psi}_k \left( -\int_{\mathcal{Y}} (c_k(x_k, y) - f_k(y)) d\gamma_k(y|x_k) \right) \right\} d\mathbb{P}_k(x_k) + m \right]. \quad (22)$$

where the inf is taken over families of measurable stochastic maps $\{\gamma_k(\cdot|x_k)\}_{[1:K]}$. Note that (22) is equal to (9) after the trivial interchange between the summation and inf operations. This completes the proof. $\qquad\square$

*Proof of Corollary 2.* Since $\text{SUOT}_{c_1} = \text{OT}_{c_1}$, $\overline{\psi}_1 = \text{Id}$ by the definition of convex conjugate. By substituting $\tilde{f}_1 := f_1 - \frac{m}{\lambda_1}$ and $\overline{\psi}_1 = \text{Id}$ into (9), we can obtain the congruence condition as follows:

$$\mathcal{L}^* = \sup_{\substack{m, f_{[1:K]} \\ \sum_{k=1}^K \lambda_k f_k \equiv m}} \inf_{\gamma_k(\cdot|x_k)} \sum_{k=1}^K \lambda_k \left[ \int_{\mathcal{X}_k} -\overline{\psi}_k \left( -\int_{\mathcal{Y}} (c_k(x_k, y) - f_k(y)) d\gamma_k(y|x_k) \right) d\mathbb{P}_k(x_k) + m \right] =$$

$$\sup_{\substack{m,f_{[1:K]} \\ \lambda_1 \tilde{f}_1 + \sum_{k=2}^{K} \lambda_k f_k \equiv 0}} \inf_{\gamma_k(\cdot|x_k) \in \mathcal{P}(\mathcal{Y})} \left[ \lambda_1 \int_{\mathcal{X}_1} \int_{\mathcal{Y}} \left( c_1(x_1,y) - \left( \tilde{f}_1(y) + \frac{m}{\lambda_1} \right) \right) d\gamma_1(y|x_1) d\mathbb{P}_1(x_1) + \right.$$

$$\left. \sum_{k=2}^{K} \lambda_k \int_{\mathcal{X}_k} -\overline{\psi}_k \left( -\int_{\mathcal{Y}} \left( c_k(x_k,y) - f_k(y) \right) d\gamma_k(y|x_k) \right) d\mathbb{P}_k(x_k) + m \right] =$$

$$\sup_{\lambda_1 \tilde{f}_1 + \sum_{k=2}^{K} \lambda_k f_k \equiv 0} \inf_{\gamma_k(\cdot|x_k) \in \mathcal{P}(\mathcal{Y})} \left[ \lambda_1 \int_{\mathcal{X}_1} \overline{\psi}_1 \left( -\int_{\mathcal{Y}} \left( c_1(x_1,y) - \tilde{f}_1(y) \right) d\gamma_1(y|x_1) \right) d\mathbb{P}_1(x_1) + \right.$$

$$\left. \sum_{k=2}^{K} \lambda_k \int_{\mathcal{X}_k} -\overline{\psi}_k \left( -\int_{\mathcal{Y}} \left( c_k(x_k,y) - f_k(y) \right) d\gamma_k(y|x_k) \right) d\mathbb{P}_k(x_k) \right].$$

$\square$

## A.2 PROOF OF THEOREM 2

Below we recall an auxiliary thereotical result which will be used in our proof of Theorem 2.

**Theorem 4** (Connection between solutions of dual OT and UOT problems (Choi et al., 2024a)). *Assume that $\overline{\psi}$ is a continuously differentiable function. Then if the optimal potential $f^*$ delivering maximum to the dual UOT problem exists, $f^*$ is a solution of the following objective*

$$OT_c(\widetilde{\mathbb{P}}, \widetilde{\mathbb{Q}}) = \sup_f \int_{\mathcal{X}} f^c(x) d\widetilde{\mathbb{P}}(x) + \int_{\mathcal{Y}} f(y) d\widetilde{\mathbb{Q}}(y)$$

*where $d\widetilde{\mathbb{P}}(x) = \nabla \overline{\psi}_1(-(f^*)^c(x)) d\mathbb{P}(x)$ and $d\widetilde{\mathbb{Q}}(y) = \nabla \overline{\psi}_2(-f^*(y)) d\mathbb{Q}(y)$.*

This Theorem highlights the intriguing connection between the solutions of the classic OT problem and its unbalanced counterpart. Note that in the case of SUOT problem, $\overline{\psi}_2(y) = y$, thus, the measure $\widetilde{\mathbb{Q}}$ reduces to $\mathbb{Q}$. Now we are ready to prove our Theorem 2.

*Proof of Theorem 2.* Below we show that optimal potentials $f^*_{[1:K]}$ delivering maximum to the dual SUOT barycenter problem (8) are also optimal for each of the underlying SUOT($\mathbb{P}_k, \mathbb{Q}$) problems. Recall that the SUOT barycenter problem (7) admits a minimizer which we denote by $\mathbb{Q}^*$. Then:

$$\sum_{k=1}^{K} \lambda_k \text{SUOT}_{c_k, \psi_k}(\mathbb{P}_k, \mathbb{Q}^*) \overset{(8)}{=} \sup_{\substack{m \in \mathbb{R}, f_{[1:K]} \in \mathcal{C}^K(\mathcal{Y}) \\ \sum_{k=1}^{K} \lambda_k f_k \equiv m}} \sum_{k=1}^{K} \lambda_k \left[ \int_{\mathcal{X}_k} -\overline{\psi}_k(-f_k^{c_k}(x_k)) d\mathbb{P}_k(x_k) + m \right] = \quad (23)$$

$$\sup_{\substack{m \in \mathbb{R}, f_{[1:K]} \in \mathcal{C}^K(\mathcal{Y}) \\ \sum_{k=1}^{K} \lambda_k f_k \equiv m}} \sum_{k=1}^{K} \lambda_k \int_{\mathcal{X}_k} -\overline{\psi}_k(-f_k^{c_k}(x_k)) d\mathbb{P}_k(x_k) + \int_{\mathcal{Y}} \underbrace{\sum_{k=1}^{K} \lambda_k f_k(y)}_{=m} d\mathbb{Q}^*(y) \leq \quad (24)$$

$$\sup_{f_{[1:K]} \in \mathcal{C}^K(\mathcal{Y})} \sum_{k=1}^{K} \lambda_k \int_{\mathcal{X}_k} -\overline{\psi}_k(-f_k^{c_k}(x_k)) d\mathbb{P}_k(x_k) + \int_{\mathcal{Y}} \sum_{k=1}^{K} \lambda_k f_k(y) d\mathbb{Q}^*(y) = \sum_{k=1}^{K} \lambda_k \text{SUOT}_{c_k, \psi_k}(\mathbb{P}_k, \mathbb{Q}^*). \quad (25)$$

Here the transition from line (23) to (24) follows from the fact that the optimization is performed over $m$-congruence potentials. If we then remove this restriction on potentials, the value of sup can become bigger which explains the transition between (24)-(25). Importantly, we get that the final objective in line (25) is equal to the initial objective in line (23). Thus, the inequality in line (24) turns into equality. At the same time, it justifies that the potentials $f^*_{[1:K]}$ delivering maximum to the SUOT barycenter (8) problem coincide with the optimal potentials for each of the SUOT problems.

Then, thanks to Theorem 4, the SUOT barycenter problem (25) admits a reformulation:

$$\mathcal{L}^* = \sum_{k=1}^{K} \lambda_k \text{SUOT}(\mathbb{P}_k, \mathbb{Q}^*) = \sum_{k=1}^{K} \lambda_k \text{OT}(\widetilde{\mathbb{P}}_k, \mathbb{Q}^*) \quad (26)$$

where each of the distributions $\widetilde{\mathbb{P}}_k$ ($k \in \overline{K}$) is specified via the optimal potential $f_k^*$ delivering maximum to the problem $\text{OT}(\widetilde{\mathbb{P}}_k, \mathbb{Q}^*)$ in its dual form (2): $d\widetilde{\mathbb{P}}_k(x_k) = \nabla\overline{\psi}(-(f_k^*)^{c_k}(x_k))d\mathbb{P}_k(x_k)$. Thus, the optimal SUOT plans now correspond to the optimal OT plans $\gamma_k^*(x, y)$ belonging to $\Pi(\widetilde{\mathbb{P}}_k, \mathbb{Q}^*)$, i.e., having marginals $(\gamma_k^*)_x = \widetilde{\mathbb{P}}_k$, $(\gamma_k^*)_y = \mathbb{Q}^*$.

Now we fix $k \in \overline{K}$. We aim to show that $\gamma_k^*(\cdot|x_k) \in \arg\inf_{\gamma_k(\cdot|x_k))} \int_{\mathcal{Y}}(c_k(x_k, y) - f_k^*(y))d\gamma(\cdot|x_k)$ where inf is taken over measurable maps $\gamma_k(\cdot|x_k)$. Then by repeating the derivations of Corollary 1 we easily get that $\gamma_k^*(\cdot|x_k) \in \arg\inf_{\gamma_k(\cdot|x_k)} \mathcal{L}(f_k^*, \gamma_k(\cdot|x_k)))$. To prove the former, we again use the fact that $\gamma_k^*$ is a solution of the balanced OT problem between the rescaled marginal $\widetilde{\mathbb{P}}_k$ and $\mathbb{Q}^*$. Note that $\nabla\overline{\psi}_k$ is a continuous function since $\overline{\psi}_k$ is assumed to be a continuously differentiable function. This yields the fact that $\widetilde{\mathbb{P}}_k$ is an absolutely continuous distribution w.r.t. $\mathbb{P}_k$. Thus, we can use (Villani et al., 2009, Remark 5.13) which states that for each $y \in \text{Supp}(\gamma_k^*(\cdot|x_k))$ it holds that $y \in \arg\inf_{y' \in \mathcal{Y}} (c_k(x_k, y') - f_k(y'))$. The latter statement is equivalent to $\gamma_k^*(\cdot|x_k) \in \arg\inf_{\gamma_k(\cdot|x_k)} \int_{\mathcal{Y}}(c_k(x_k, y) - f_k^*(y))d\gamma(\cdot|x_k)$ which completes the proof. Note that since the infimum is attained at least for one map $\gamma_k^*(\cdot|x_k)$, the $\arg\inf_{\gamma_k(\cdot|x_k)}$ in the latter equation is actually $\arg\min_{\gamma_k(\cdot|x_k)}$.

$\square$

### A.3 PROOF OF THEOREM 3

The theorem provides duality gap analysis. We are given approximations: $\widehat{m}$; $\widehat{f}_{[1:K]}$ which are $\widehat{m}$-congruent; $\widehat{\gamma}_{[1:K]}$. Recall that the gaps are defined as follows:

$$\delta_1(\widehat{f}_{[1:K]}, \widehat{\gamma}_{[1:K]}) = \widetilde{\mathcal{L}}(\widehat{f}_{[1:K]}, \widehat{\gamma}_{[1:K]}, \widehat{m}) - \inf_{\gamma_k(\cdot|x_k)\in\mathcal{P}(\mathcal{Y})} \widetilde{\mathcal{L}}(\widehat{f}_{[1:K]}, \gamma_{[1:K]}, \widehat{m}); \quad (27)$$

$$\delta_2(\widehat{f}_{[1:K]}) = \mathcal{L}^* - \inf_{\gamma_k(\cdot|x_k)\in\mathcal{P}(\mathcal{Y})} \widetilde{\mathcal{L}}(\widehat{f}_{[1:K]}, \gamma_{[1:K]}, \widehat{m}). \quad (28)$$

*Proof of Theorem 3.* Our proof considerably extends the proof of (Kolesov et al., 2024a, Theorem 4.2) for the case of the classical cost function.

**Part 1**. Several notations.

To begin with, we introduce several notations ($\mathbb{Q}^*$ is a SUOT barycenter):

$$\mathcal{V}_k(f_k, \gamma_k) \stackrel{\text{def}}{=} \int_{\mathcal{X}_k} -\overline{\psi}_k\left(-\int_{\mathcal{Y}}(c_k(x_k, y) - f_k(y))d\gamma_k(y|x_k)\right)d\mathbb{P}_k(x_k); \quad (29)$$

$$\widetilde{\mathcal{V}}_k(f_k, \gamma_k) \stackrel{\text{def}}{=} \mathcal{V}_k(f_k, \gamma_k) + \int_{\mathcal{Y}} f_k(y)d\mathbb{Q}^*(y); \quad (30)$$

$$\mathcal{J}_k(f_k) \stackrel{\text{def}}{=} \inf_{\gamma_k(\cdot|x_k)\in\mathcal{P}(\mathcal{Y})} \mathcal{V}_k(f_k, \gamma_k); \quad (31)$$

$$\widetilde{\mathcal{J}}_k(f_k) \stackrel{\text{def}}{=} \mathcal{J}_k(f_k) + \int_{\mathcal{Y}} f_k(y)d\mathbb{Q}^*(y); \quad (32)$$

Note that the following holds:

$$\mathcal{L}^* = \sup_{\substack{m\in\mathbb{R}, \\ f_{[1:K]}\in\mathcal{C}^K(\mathcal{Y}) \\ \sum_{k=1}^K \lambda_k f_k \equiv m}} \left\{\sum_{k=1}^K \lambda_k \mathcal{J}_k(f_k) + m\right\} = \sup_{\substack{m\in\mathbb{R}, \\ f_{[1:K]}\in\mathcal{C}^K(\mathcal{Y}) \\ \sum_{k=1}^K \lambda_k f_k \equiv m}} \inf_{\gamma(\cdot|x_k)\in\mathcal{P}(\mathcal{Y})} \left\{\sum_{k=1}^K \lambda_k \mathcal{V}_k(f_k, \gamma_k) + m\right\}.$$

Also, if $f_{[1:K]}$ are $m$-congruent, then $\sum_{k=1}^K \lambda_k \int_{\mathcal{Y}} f_k(y)d\mathbb{Q}^*(y) = m$ and therefore:

$$\sum_{k=1}^K \lambda_k \widetilde{\mathcal{V}}(f_f, \gamma_k) = \sum_{k=1}^K \lambda_k \mathcal{V}_k(f_k, \gamma_k) + m = \widetilde{\mathcal{L}}(f_{[1:K]}, \gamma_{[1:K]}, m);$$

$$\sum_{k=1}^{K} \lambda_k \widetilde{\mathcal{J}}_k(f_k) = \sum_{k=1}^{K} \lambda_k \mathcal{J}_k(f_k) + m = \inf_{\gamma_k(\cdot|x_k)\in\mathcal{P}(\mathcal{Y})} \widetilde{\mathcal{L}}(f_{[1:K]}, \gamma_{[1:K]}, m). \tag{33}$$

Now we start the proof.

**Part 2**. Analysis of gap $\delta_1$.

At first, we express gap $\delta_1$ in terms of newly introduced $\mathcal{V}_k$ (29), $\mathcal{J}_k$ (31) as follows:

$$
\begin{aligned}
\delta_1 &= \widetilde{\mathcal{L}}(\widehat{f}_{[1:K]}, \widehat{\gamma}_{[1:K]}, \widehat{m}) - \inf_{\gamma_k(\cdot|x_k)\in\mathcal{P}(\mathcal{Y})} \widetilde{\mathcal{L}}(\widehat{f}_{[1:K]}, \gamma_{[1:K]}, \widehat{m}) \\
&= \sum_{k=1}^{K} \lambda_k \{ \mathcal{V}_k(\widehat{f}_k, \widehat{\gamma}_k) - \mathcal{J}_k(\widehat{f}_k) \} \\
&= \sum_{k=1}^{K} \lambda_k \delta_{1,k}(\widehat{f}_k, \widehat{\gamma}_k),
\end{aligned}
$$

$$\text{where} \quad \delta_{1,k}(\widehat{f}_k, \widehat{\gamma}_k) \stackrel{\text{def}}{=} \mathcal{V}_k(\widehat{f}_k, \widehat{\gamma}_k) - \mathcal{J}_k(\widehat{f}_k); \tag{34}$$

We analyze gaps $\delta_{1,k}$ for each particular $k \in [1, K]$ separately. For convenience, we introduce the function $\widehat{g}_k(x_k, y) \stackrel{\text{def}}{=} c_k(x_k, y) - \widehat{f}_k(y)$, $x_k \in \mathcal{X}_k$, $y \in \mathcal{Y}$. Note that $\widehat{g}_k$ is $\beta$-strongly convex in the second argument by the Theorem assumption. Define the maps $T_k^{\widehat{f}} : \mathcal{X}_k \to \mathcal{Y}$ as follows:

$$T_k^{\widehat{f}}(x_k) \stackrel{\text{def}}{=} \arg\inf_{y\in\mathcal{Y}} \widehat{g}_k(x_k, y).$$

Note that $T_k^{\widehat{f}}$ is correctly defined (since $\widehat{g}_k$ is strongly convex) and Borel (cf. the proof of (Kolesov et al., 2024a, Theorem 4.2). It induces the family of conditional distributions:

$$
\begin{aligned}
x_k &\mapsto \gamma_k^{\widehat{f}}(\cdot|x_k) \in \mathcal{P}(\mathcal{Y}); \\
\gamma_k^{\widehat{f}}(\cdot|x_k) &\stackrel{\text{def}}{=} \delta_{T_k^{\widehat{f}}(x_k)}(\cdot).
\end{aligned} \tag{35}
$$

For any family $\gamma_k(\cdot|x_k) \in \mathcal{P}(\mathcal{Y})$ we have:

$$
\begin{aligned}
\mathcal{V}_k(\widehat{f}_k, \gamma_k) &= \int_{\mathcal{X}_k} -\overline{\psi}_k\Big(-\int_{\mathcal{Y}} \widehat{g}_k(x_k, y) d\gamma_k(y|x_k)\Big) d\mathbb{P}_k(x_k) \tag{36} \\
&\geq \int_{\mathcal{X}_k} -\overline{\psi}_k\Big(-\widehat{g}_k(x_k, T_k^{\widehat{f}}(x_k))\Big) d\mathbb{P}_k(x_k) \tag{37} \\
&= \int_{\mathcal{X}_k} -\overline{\psi}_k\Big(-\int_{\mathcal{Y}} \widehat{g}_k(x_k, y) d\gamma_k^{\widehat{f}}(y|x_k)\Big) d\mathbb{P}_k(x_k) = \mathcal{V}_k(\widehat{f}_k, \gamma_k^{\widehat{f}}),
\end{aligned}
$$

where in transition from (36) to (37) we utilize the properties of $\overline{\psi}_k$ and $T_k^{\widehat{f}}$:

$$
\begin{aligned}
\int_{\mathcal{Y}} \widehat{g}_k(x_k, y) d\gamma_k(y|x_k) &\geq \widehat{g}_k(x_k, T_k^{\widehat{f}}(x_k)) \Rightarrow \\
-\int_{\mathcal{Y}} \widehat{g}_k(x_k, y) d\gamma_k(y|x_k) &\leq -\widehat{g}_k(x_k, T_k^{\widehat{f}}(x_k)) \overset{\overline{\psi}_k \text{ is non-decreasing}}{\Rightarrow} \\
\overline{\psi}_k\Big(-\int_{\mathcal{Y}} \widehat{g}_k(x_k, y) d\gamma_k(y|x_k)\Big) &\leq \overline{\psi}_k\Big(-\widehat{g}_k(x_k, T_k^{\widehat{f}}(x_k))\Big) \Rightarrow
\end{aligned}
$$

$$-\overline{\psi}_k\Big(-\int_{\mathcal{Y}}\widehat{g}_k(x_k,y)d\gamma_k(y|x_k)\Big) \geq -\overline{\psi}_k\Big(-\widehat{g}_k(x_k,T_k^{\widehat{f}}(x_k))\Big).$$

From the above, we conclude that $\gamma_k^{\widehat{f}}$ minimizes the functional $\gamma_k \mapsto \mathcal{V}_k(\widehat{f}_k,\gamma_k)$. In particular, it delivers the value of $\mathcal{J}_k(\widehat{f}_k)$:

$$\mathcal{J}_k(\widehat{f}_k) = \mathcal{V}_k(\widehat{f}_k,\gamma_k^{\widehat{f}}). \tag{38}$$

From the perspective of (38), gap $\delta_{1,k}$ could be analyzed as follows:

$$\delta_{1,k}(\widehat{f}_k,\widehat{\gamma}_k) = \mathcal{V}_k(\widehat{f}_k,\widehat{\gamma}_k) - \mathcal{V}_k(\widehat{f}_k,\gamma_k^{\widehat{f}})$$

$$= \int_{\mathcal{X}_k}\Big\{\overline{\psi}_k\Big(-\widehat{g}_k(x_k,T_k^{\widehat{f}}(x_k))\Big) - \overline{\psi}_k\Big(-\int_{\mathcal{Y}}\widehat{g}_k(x_k,y)d\widehat{\gamma}_k(y|x_k)\Big)\Big\}d\mathbb{P}_k(x_k) \tag{39}$$

$$\geq \eta\int_{\mathcal{X}_k}\Big\{\int_{\mathcal{Y}}\widehat{g}_k(x_k,y)d\widehat{\gamma}_k(y|x_k) - \widehat{g}_k(x_k,T_k^{\widehat{f}}(x_k))\Big\}d\mathbb{P}_k(x_k) \tag{40}$$

$$= \eta\int_{\mathcal{X}_k}\int_{\mathcal{Y}}\Big\{\widehat{g}_k(x_k,y) - \widehat{g}_k(x_k,T_k^{\widehat{f}}(x_k))\Big\}d\widehat{\gamma}_k(y|x_k)d\mathbb{P}_k(x_k) \tag{41}$$

$$\geq \frac{\eta\beta}{2}\int_{\mathcal{X}_k}\int_{\mathcal{Y}}\|y - T_k^{\widehat{f}}(x_k)\|_2^2 d\widehat{\gamma}_k(y|x_k)d\mathbb{P}_k(x_k). \tag{42}$$

*Transition from* (39) *to* (40). Note that:

$$-\widehat{g}_k(x_k,T_k^{\widehat{f}}(x_k)) \geq -\underbrace{\int_{\mathcal{Y}}\widehat{g}_k(x_k,y)d\widehat{\gamma}_k(y|x_k)}_{\overset{\text{def}}{=}\widehat{g}_k^{\gamma}} \geq -\max_{x_k\in\mathcal{X}_k; y\in\mathcal{Y}}\widehat{g}_k(x_k,y) \geq -b.$$

The first-order properties of convex differentiable function $\overline{\psi}_k$ read as:

$$\nabla\overline{\psi}_k\big(-\widehat{g}_k^{\gamma}\big) \geq \nabla\overline{\psi}_k(-b) = \eta;$$

$$\overline{\psi}_k\big(-\widehat{g}_k(x_k,T_k^{\widehat{f}}(x_k))\big) \geq \overline{\psi}_k\big(-\widehat{g}_k^{\gamma}\big) + \nabla\overline{\psi}_k(-\widehat{g}_k^{\gamma})\big[\widehat{g}_k^{\gamma} - \widehat{g}_k(x_k,T_k^{\widehat{f}}(x_k))\big]$$

Based on them, we derive:

$$\overline{\psi}_k\big(-\widehat{g}_k(x_k,T_k^{\widehat{f}}(x_k))\big) - \overline{\psi}_k\big(-\widehat{g}_k^{\gamma}\big) \geq \eta\big[\widehat{g}_k^{\gamma} - \widehat{g}_k(x_k,T_k^{\widehat{f}}(x_k))\big],$$

which validates the transition from from (39) to (40).

*Transition from* (41) *to* (42). Note that $T_k^{\widehat{f}}(x_k)$ is the minimizer of $y \mapsto \widehat{g}_k(x_k,y)$. Therefore, due to $\beta$ - strong convexity of $\widehat{g}_k$ on the second argument it holds:

$$\frac{\beta}{2}\|T_k^{\widehat{f}}(x_k) - y\|_2^2 \leq \widehat{g}_k(x_k,y) - \widehat{g}_k(x_k,T_k^{\widehat{f}}(x_k)),$$

which validates the transition.

**Part 3**. Analysis of gap $\delta_2$.

Note that we work under the conditions of Theorem 2. In particular, we assume that there exist $m^*$ and $m^*$-congruent potentials $\{f_k^* \in \mathcal{C}(\mathcal{Y})\}_{k=1}^K$ which deliver maximum to (9). Also, we denote $\{\gamma_k^*\}_{k=1}^K$ as the family of optimal SUOT plans between $\mathbb{P}_k$ and $\mathbb{Q}^*$.

Under these assumptions, we note that:

$$\mathcal{L}^* \overset{\text{cf. (8)}}{=} \sum_{k=1}^K \lambda_k\Big[-\overline{\psi}_k\Big(-(f_k^*)^{c_k}(x_k)\Big)d\mathbb{P}_k(x_k) + m^*\Big] \tag{43}$$

$$= \sum_{k=1}^{K} \lambda_k \Big[ -\overline{\psi}_k \Big( -\int_{\mathcal{Y}} \big( c_k(x_k, y) - f_k^*(y) \big) d\gamma_k^*(y|x_k) \Big) d\mathbb{P}_k(x_k) + m^* \Big], \quad (44)$$

$$= \sum_{k=1}^{K} \lambda_k \widetilde{\mathcal{V}}_k(f_k^*, \gamma_k^*) \quad (45)$$

where the transition from (43) to (44) follows from the proof of Theorem 2. In particular,

$$\int_{\mathcal{Y}} \big( c_k(x_k, y) - f_k^*(y) \big) d\gamma_k^*(y|x_k) = (f_k^*)^{c_k}(x_k).$$

Thanks to (45) and (33), one can decompose gap $\delta_2$ as follows:

$$\begin{aligned}
\delta_2 &= \mathcal{L}^* - \inf_{\gamma_k(\cdot|x_k) \in \mathcal{P}(\mathcal{Y})} \widetilde{\mathcal{L}}(\widehat{f}_{[1:K]}, \gamma_{[1:K]}, \widehat{m}) \\
&= \sum_{k=1}^{K} \lambda_k \big( \widetilde{\mathcal{V}}_k(f_k^*, \gamma_k^*) - \widetilde{\mathcal{J}}_k(\widehat{f}_k) \big) \\
&\stackrel{\text{cf. (38)}}{=} \sum_{k=1}^{K} \lambda_k \big( \widetilde{\mathcal{V}}_k(f_k^*, \gamma_k^*) - \widetilde{\mathcal{V}}_k(\widehat{f}_k, \gamma_k^{\widehat{f}}) \big) \\
&= \sum_{k=1}^{K} \lambda_k \delta_{2,k}(\widehat{f}_k),
\end{aligned}$$

$$\text{where} \quad \delta_{2,k}(\widehat{f}_k) \stackrel{\text{def}}{=} \widetilde{\mathcal{V}}_k(f_k^*, \gamma_k^*) - \widetilde{\mathcal{V}}_k(\widehat{f}_k, \gamma_k^{\widehat{f}}).$$

In what follows, we analyze gap $\delta_{2,k}$ for each particular $k \in [1:K]$:

$$\delta_{2,k}(\widehat{f}_k) = \int_{\mathcal{X}_k} \Big\{ \overline{\psi}_k \Big( -\widehat{g}_k(x_k, T_k^{\widehat{f}}(x_k)) \Big) - \overline{\psi}_k \Big( -(f_k^*)^{c_k}(x_k) \Big) \Big\} d\mathbb{P}_k(x_k) \quad (46)$$

$$+ \int_{\mathcal{Y}} \big\{ f_k^*(y) - \widehat{f}_k(y) \big\} d\mathbb{Q}^*(y)$$

$$\geq \int_{\mathcal{X}_k} \Big\{ (f_k^*)^{c_k}(x_k) - \widehat{g}_k(x_k, T_k^{\widehat{f}}(x_k)) \Big\} \underbrace{\nabla \overline{\psi}_k \big( -(f_k^*)^{c_k}(x_k) \big) d\mathbb{P}_k(x_k)}_{= d(\gamma_k^*)_x(x_k) \,,\, \text{cf. (11)}} \quad (47)$$

$$+ \int_{\mathcal{Y}} \big\{ f_k^*(y) - \widehat{f}_k(y) \big\} d\mathbb{Q}^*(y)$$

$$= \int_{\mathcal{X}_k} \int_{\mathcal{Y}} \big\{ c_k(x_k, y) - f_k^*(y) \big\} \underbrace{d\gamma_k^*(y|x_k) d(\gamma_k^*)_x(x_k)}_{= d\gamma_k^*(x_k, y)} + \int_{\mathcal{Y}} f_k^*(y) d\mathbb{Q}^*(y)$$

$$- \int_{\mathcal{X}_k} \widehat{g}_k(x_k, T_k^{\widehat{f}}(x_k)) d(\gamma_k^*)_x(x_k) - \int_{\mathcal{Y}} \widehat{f}_k(y) d\mathbb{Q}^*(y)$$

$$= \int_{\mathcal{X}_k} \int_{\mathcal{Y}} \big\{ c_k(x_k, y) - \cancel{f_k^*(y)} \big\} d\gamma_k^*(x_k, y) + \cancel{\int_{\mathcal{X}_k} \int_{\mathcal{Y}} f_k^*(y) d\gamma_k^*(x_k, y)}$$

$$- \int_{\mathcal{X}_k} \int_{\mathcal{Y}} \widehat{g}_k(x_k, T_k^{\widehat{f}}(x_k)) d\gamma_k^*(x_k, y) - \int_{\mathcal{X}_k} \int_{\mathcal{Y}} \widehat{f}_k(y) d\gamma_k^*(x_k, y)$$

$$= \int_{\mathcal{X}_k} \int_{\mathcal{Y}} \big\{ \widehat{g}_k(x_k, y) - \widehat{g}_k(x_k, T_k^{\widehat{f}}(x_k)) \big\} d\gamma_k^*(x_k, y)$$

$$= \int_{\mathcal{X}_k} \Big\{ \int_{\mathcal{Y}} \widehat{g}_k(x_k, y) d\gamma_k^*(y|x_k) - \widehat{g}_k(x_k, T_k^{\widehat{f}}(x_k)) \Big\} \nabla \overline{\psi}_k \big( -(f_k^*)^{c_k}(x_k) \big) d\mathbb{P}_k(x_k) \quad (48)$$

$$\geq \quad \eta \int\limits_{\mathcal{X}_k} \left\{ \int\limits_{\mathcal{Y}} \widehat{g}_k(x_k, y) d\gamma_k^*(y|x_k) - \widehat{g}_k(x_k, T_k^{\widehat{f}}(x_k)) \right\} d\mathbb{P}_k(x_k) \tag{49}$$

$$= \quad \eta \int\limits_{\mathcal{X}_k} \int\limits_{\mathcal{Y}} \left\{ \widehat{g}_k(x_k, y) - \widehat{g}_k(x_k, T_k^{\widehat{f}}(x_k)) \right\} d\gamma_k^*(y|x_k) d\mathbb{P}_k(x_k)$$

$$\geq \quad \frac{\eta\beta}{2} \int\limits_{\mathcal{X}_k} \int\limits_{\mathcal{Y}} \|y - T_k^{\widehat{f}}(x_k)\|_2^2 d\gamma_k^*(y|x_k) d\mathbb{P}_k(x_k) \tag{50}$$

*Transition from* (46) *to* (47) follows from the first-order property of convex differentiable function $\overline{\psi}_k$:

$$\overline{\psi}_k\big(-\widehat{g}_k(x_k, T_k^{\widehat{f}}(x_k))\big) \geq \overline{\psi}_k\big(-(f_k^*)^{c_k}(x_k)\big) + \nabla\overline{\psi}_k\big(-(f_k^*)^{c_k}(x_k)\big)\big[(f_k^*)^{c_k}(x_k) - \widehat{g}_k(x_k, T_k^{\widehat{f}}(x_k))\big]$$

*Transition from* (48) *to* (49). Since $T_k^{\widehat{f}}$ minimizes $y \mapsto \widehat{g}_k(x_k, y)$ then:

$$\int\limits_{\mathcal{Y}} \widehat{g}_k(x_k, y) d\gamma_k^*(y|x_k) - \widehat{g}_k(x_k, T_k^{\widehat{f}}(x_k)) \geq 0. \tag{51}$$

Also note that:

$$-(f_k^*)^{c_k}(x_k) = -\int\limits_{\mathcal{Y}} \big(c_k(x_k, y) - f_k^*(y)\big) d\gamma_k^*(y|x_k) \geq -\max_{x_k \in \mathcal{X}_k, y \in \mathcal{Y}} \big\{ c_k(x_k, y) - f_k^*(y) \big\} \geq -b,$$

and, therefore,

$$\nabla\overline{\psi}_k\big(-(f_k^*)^{c_k}(x_k)\big) \geq \nabla\overline{\psi}_k(-b) = \eta. \tag{52}$$

The combination of (51) and (52) validates the transition.

**Part 4**. Bringing it all together.

Summarizing inequalities for $\delta_{1,k}$ (42) and $\delta_{2,k}$ (50) yields:

$$\delta_{1,k}(\widehat{f}_k, \widehat{\gamma}_k) + \delta_{2,k}(\widehat{f}_k)$$
$$\geq \quad \frac{\eta\beta}{2} \left\{ \int\limits_{\mathcal{X}_k \times \mathcal{Y}} \|y - T_k^{\widehat{f}}(x_k)\|_2^2 d\widehat{\gamma}_k(y|x_k) d\mathbb{P}_k(x_k) + \int\limits_{\mathcal{X}_k \times \mathcal{Y}} \|y - T_k^{\widehat{f}}(x_k)\|_2^2 d\gamma_k^*(y|x_k) d\mathbb{P}_k(x_k) \right\}$$
$$\geq \quad \frac{\eta\beta}{2} \int\limits_{\mathcal{X}_k} \mathbb{W}_2^2\big(\widehat{\gamma}_k(\cdot|x_k), \gamma_k^*(\cdot|x_k)\big) d\mathbb{P}_k(x_k),$$

where the last inequality is obtained analogously to the similar one in the proof of (Kolesov et al., 2024a, Theorem 4.2).

Aggregating the inequalities for $\delta_{1,k} + \delta_{2,k}$ with weights $\lambda_k$ finishes the proof of the Theorem:

$$\delta_1 + \delta_2 = \sum_{k=1}^K \lambda_k(\delta_{1,k} + \delta_{2,k}) \geq \frac{\eta\beta}{2} \sum_{k=1}^K \lambda_k \int_{\mathcal{X}_k} \mathbb{W}_2^2\big(\widehat{\gamma}_k(\cdot|x), \gamma_k^*(\cdot|x)\big) d\mathbb{P}_k(x).$$

$\square$

# B   EXTENDED DISCUSSION OF RELATED WORKS

Below we give an overview of the continuous OT solvers with a specific focus on unbalanced setup.

**Neural OT/UOT solvers.** Neural network-based continuous OT is a popular and fruitful area of recent generative modelling research. Some keynote solvers include: (Makkuva et al., 2020; Korotin et al., 2021a; Amos, 2023) (ICNN-based, quadratic cost); (Seguy et al., 2018; Daniels et al., 2021; Mokrov et al., 2024) (Entropic OT); (Vargas et al., 2021; De Bortoli et al., 2021; Gushchin et al.,

2023; Tong et al., 2023; Shi et al., 2024; Gushchin et al., 2024b; Korotin et al., 2024; Gushchin et al., 2024a), (Schrödinger bridge); (Liu et al., 2023; Tong et al., 2024; Kornilov et al., 2024; Klein et al., 2024) (Flow matching). Of special importance for our developed method are max-min (adversarial) OT solvers based on (semi-) dual OT formulation (Rout et al., 2022; Korotin et al., 2023b;a; Fan et al., 2023; Asadulaev et al., 2024; Korotin et al., 2022b; 2021b). Recently, the adversarial methodology has been extended to unbalanced setup (Choi et al., 2024a; Gazdieva et al., 2023; Choi et al., 2024b), opening up a new intriguing research direction in the field of robust continuous OT.

## C  IMPLEMENTATION DETAILS

In this section, we provide additional details about experiments in our paper.

### C.1  DESCRIPTION OF TOY EXPERIMENTS

**Moon, Spiral, and 8-Gaussian Datasets.** The datasets used in Section 5.1 were implemented by following (Choi et al., 2024b).

**Datasets for Class Imbalance Experiments.** For the distribution $\mathbb{P}_1$ and $\mathbb{P}_2$, we employ the Gaussian mixture of $\frac{1}{4}\mathcal{N}((-5,4), 0.4^2) + \frac{3}{4}\mathcal{N}((-5,-4), 0.4^2)$ and $\frac{3}{4}\mathcal{N}((5,4), 0.4^2) + \frac{1}{4}\mathcal{N}((5,-4), 0.4^2)$, respectively.

**Datasets for Outlier Experiments.** We consider three marginal distributions, denoted as $\mathbb{P}_1$, $\mathbb{P}_2$, and $\mathbb{P}_3$. For $\mathbb{P}_1$ and $\mathbb{P}_2$, we generate datasets consisting of 95% in-distribution data and 5% outliers. In contrast, $\mathbb{P}_3$ consists solely of in-distribution data. The in-distribution data for each marginal follows a Gaussian mixture model with four modes and uniform weights, i.e., $\sum_{k=1}^{4} \frac{1}{4}\mathcal{N}(m_k, \sigma^2 I)$. The means and variances for $\mathbb{P}_1$, $\mathbb{P}_2$, and $\mathbb{P}_3$ are defined as follows:

- $\mathbb{P}_1$: $m_1 = (-5,-1)$, $m_2 = (5,1)$, $m_3 = (1,-5)$, $m_4 = (-1,5)$, with $\sigma = 0.1$,
- $\mathbb{P}_2$: $m_1 = (-5,1)$, $m_2 = (5,-1)$, $m_3 = (1,5)$, $m_4 = (-1,-5)$, with $\sigma = 0.1$,
- $\mathbb{P}_3$: $m_1 = (-5,0)$, $m_2 = (5,0)$, $m_3 = (0,5)$, $m_4 = (0,-5)$, with $\sigma = 0.1$,

respectively. For the outlier distributions of $\mathbb{P}_1$ and $\mathbb{P}_2$, we again use a Gaussian mixture model with four modes, $\sum_{k=1}^{4} \frac{1}{4}\mathcal{N}(m_k, \sigma^2 I)$, with the means and variances defined as:

- $\mathbb{P}_1$ outliers: $m_1 = (10,2)$, $m_2 = (10,1)$, $m_3 = (10,0)$, $m_4 = (10,-1)$, with $\sigma = 0.02$,
- $\mathbb{P}_2$ outliers: $m_1 = (-10,1)$, $m_2 = (-10,0)$, $m_3 = (10,-1)$, $m_4 = (10,-2)$, with $\sigma = 0.02$,

respectively.

**Implementation Details.** For the experiments in Section 5.1 and the class imbalanced experiments, we use the number of iterations of 10K. For the outlier experiments in Section 5.2, we employ the number of iterations of 20K. We use the batch size of 1024. We update $m$ with the Adam Optimizer with learning rate of $10^{-3}$ and $(\beta_1, \beta_2) = (0, 0.9)$. For other hyperparameters, we follow the experimental settings of (Kolesov et al., 2024a).

**Evaluation metric.** As discussed in Section 5.1, we utilize the $\mathbb{W}_2$ and $\mathcal{L}_2$ metrics, defined as $\mathbb{W}_2^2(T_{1\#}\mathbb{P}_1, \mathbb{Q}^*)$ and $\|T_1 - T_1^*\|_{L^2(\mathbb{P}_1)}^2$, respectively. Here, $\mathbb{Q}^*$ and $T_1^*$ represent the true barycenter and transport maps. To approximate $(T_1^*, \mathbb{Q}^*)$, we solve the discrete Unbalanced Barycenter problem. Specifically, we sample 2000 points and solve the Unbalanced Optimal Transport (UOT) problem from $\mathbb{P}_1$ to $\mathbb{P}_2$ using the Python Optimal Transport (POT) library (Flamary et al., 2021). The obtained discrete UOT map $T_{uot}^*$ is then used to define $T_1^* = \lambda_1 \mathrm{Id} + \lambda_2 T_{uot}^*$, and $\mathbb{Q}^* = T_{1\#}^*\mathbb{P}_1$. Using these discrete samples and transport maps, we compute the approximation of the metrics.

**Training/Inference Time.** The training and inference times of our model, along with comparisons to baseline methods, are presented in Table 2 and Table 3. In the case of toy experiments, the training time of U-NOTB is comparable to that of NOTB (Kolesov et al., 2024a). The inference time of U-NOTB is typically 2-10 times slower than NOTB. This gap is due to the additional computational complexity introduced by rejection sampling, which requires calculating the $c$-transform of the

potential function $\widehat{f}_k$, i.e., $\widehat{f}_k^c(x) \approx c(x, \widehat{T}_k(x)) - \widehat{f}(\widehat{T}_k(x))$. Thus, this process involves a forward pass through the learned potential function. Moreover, it requires to compute the cost functional, which is an additional burden for StyleGAN experiments. However, we would like to highlight that thanks to our proposed sampling method, our solver gains robustness to outliers and class imbalancedness.

| Experiment | 2D Toy (§5.1) | Class Imbalance (§5.2) | Outlier (§5.2) | Shape-Color (§5.3) |
|---|---|---|---|---|
| UOTM | 4-5 mins | - | - | - |
| Minibatch UOT | 2h | - | - | - |
| NOTB | 6-7 mins | 6-7 mins | 42-45 mins | - |
| U-NOTB | 6-7 mins | 6-7 mins | 42-45 mins | 3h 20 mins |
| GPU | RTX 2080Ti | RTX 2080Ti | RTX 2080Ti | RTX 3090Ti |

Table 2: Training time for various experiments.

| Experiment | 2D Toy (§5.1) | Class Imbalance (§5.2) | Outlier (§5.2) | Shape-Color (§5.3) |
|---|---|---|---|---|
| UOTM | 0.001 sec | - | - | - |
| Minibatch UOT | 0.001 sec | - | - | - |
| NOTB | 0.001 sec | 0.001 sec | 0.001 sec | 0.02 sec |
| U-NOTB | 0.001 sec | 0.003 sec | 0.003 sec | 0.18 sec |
| GPU | RTX 2080Ti | RTX 2080Ti | RTX 2080Ti | RTX 3090Ti |

Table 3: Inference time for various experiments. We report the inference time of 1000 samples for synthetic experiments (§5.1, §5.2). For StyleGAN experiment (§5.3), we evaluated with 220 samples.

## C.2 STYLEGAN EXPERIMENTS

Unless otherwise stated, we follow the implementation of (Kolesov et al., 2024a). Note that we use Adam Optimizers with $(\beta_1, \beta_2) = (0, 0.9)$.

The exact hyperparameters for all experiments are described in Table 4.

| Experiment | 2D Toy (§5.1) | Class Imbalance (§5.2) | Outlier (§5.2) | Shape-Color (§5.3) |
|---|---|---|---|---|
| D | 2 | 2 | 2 | 512 |
| K | 2 | 2 | 3 | 2 |
| $\tau$ | 5 | 1,20,200 | 1,20,200 | 10 |
| batch size | 1024 | 256 | 256 | 64 |
| Epochs | 10K | 10K | 20K | 3K |
| $N_T$ | 3 | 3 | 5 | 10 |
| $\lambda_1$ | 1/2 | 1/2 | 1/3 | 1/2 |
| $\lambda_2$ | 1/2 | 1/2 | 1/3 | 1/3 |
| $\lambda_3$ | - | - | 1/3 | - |
| Divergence $\psi_1$ | balanced | KL | KL | Softplus |
| Divergence $\psi_2$ | KL | KL | KL | Softplus |
| Divergence $\psi_3$ | - | - | KL | - |
| $f_{k,\theta}$ | MLP | MLP | MLP | ResNet |
| $T_{k,\omega}$ | MLP | MLP | MLP | ResNet |
| $lr_{f_{k,\theta}}$ | 1e-3 | 1e-3 | 1e-3 | 1e-4 |
| $lr_{T_{k,\omega}}$ | 1e-3 | 1e-3 | 1e-3 | 1e-7 |
| $lr_m$ | 1e-3 | 1e-3 | 1e-3 | 1e-2 |

Table 4: Hyper-parameter settings of Algorithm 1 for various experiments.

## D EXTENDED EXPERIMENTS

In this section, we provide additional experimental results. In Appendices D.1 and D.2, we demonstrate the performance of our solver and baselines in OT/SUOT barycenter problem for Gaussian distributions with computable ground-truth solutions. In Appendix D.3, we conduct high-dimensional

experiment demonstrating the practical advantages of our solver, i.e., its ability to manipulate images through interpolating image distributions on the image manifolds.

## D.1 BALANCED OT BARYCENTERS FOR GAUSSIAN DISTRIBUTIONS

In the balanced OT barycenter problem for Gaussian distributions, the ground-truth barycenter is known to be Gaussian and can be estimated using the fixed point iteration procedure (Álvarez-Esteban et al., 2016). In this section, we tested our solver in this balanced setup keeping in mind that for large unbalancedness parameters $\tau$, it should provide a good approximation of balanced OT barycenter problem solutions. **Ultimately, we show that while our solver is designed to tackle an _unbalanced_ OT barycenter problem, its performance in the different balanced OT barycenter problem is comparable to the current SOTA solvers.**

We consider $K = 3$ Gaussian input distributions with weights $\lambda_1 = \lambda_2 = 0.25$, $\lambda_3 = 0.5$, and quadratic OT cost functions $c_k(x, y) = \frac{\|x_k - y\|^2}{2}$ following the experimental setup which was firstly introduced in (Kolesov et al., 2024b) and also used in (Kolesov et al., 2024a, Appendix B.1). We perform comparison with three recent approaches for continuous barycenter estimation $-$ (Kolesov et al., 2024a, NOTB), (Kolesov et al., 2024b, EgBary) and (Korotin et al., 2022a, WIN). For EgBary approach, which solves the Entropy-resugularized OT (EOT) barycenter problem, we consider small entropy regularization parameter $\varepsilon = 0.01$. For completeness, we also tested the performance of the classic approach (Cuturi & Doucet, 2014, FCWB) which approximates the barycenter by a discrete distribution on a fixed number of free-support points.

For our U-NOTB solver, we consider KL divergencies and varying unbalancedness parameter $\tau \in [1, 10^1, 10^2, 10^3, 10^4, 10^5]$ expecting that for large $\tau$, our solver will provide good results in solving the balanced problem. We assessed the performance of solvers using the weighted unexplained variance percentage metrics $\mathcal{L}_2\text{-UVP}(\hat{T}) = 100 \cdot [\frac{\|\hat{T} - T^*\|^2_{L^2(\mathbb{P})}}{\text{var}(\mathbb{Q}^*)}]\%$ where $\mathbb{Q}^*$ is a given ground-truth OT barycenter. For the continuous baseline solvers (NOTB, EgBary, WIN), we report the results given in (Kolesov et al., 2024a, Table 3). Specifically, for our solver and EgBary which learn the optimal plans, we consider their barycentric projections. For the assessment of FCWB, we first calculate the discrete barycenter using large number of samples from input distributions $\mathbb{P}_k$. Then we compute the optimal plans between the input and this barycenter distributions (Flamary et al., 2021), and consider their barycentric projections. Table 5 presents the weighted sum of $\mathcal{L}_2$-UVP values w.r.t. the barycenter weights $\lambda_k$.

| Method/Dim | 2 | 4 | 8 | 16 | 64 |
|---|---|---|---|---|---|
| **Ours** ($\tau = 1$) | 3.10 | 2.39 | 1.64 | 1.44 | 1.39 |
| **Ours** ($\tau = 10^1$) | 0.03 | 0.09 | 0.06 | 0.08 | 0.09 |
| **Ours** ($\tau = 10^2$) | 0.02 | 0.03 | **0.03** | 0.05 | **0.07** |
| **Ours** ($\tau = 10^3$) | 0.02 | 0.03 | 0.04 | 0.05 | 0.08 |
| **Ours** ($\tau = 10^4$) | 0.02 | 0.03 | 0.04 | 0.05 | 0.08 |
| NOTB | **0.01** | **0.02** | 0.04 | **0.04** | 0.08 |
| EgBary | 0.02 | 0.05 | 0.06 | 0.09 | 0.84 |
| WIN | 0.03 | 0.08 | 0.13 | 0.25 | 0.75 |
| FCWB | 2.17 | 6.51 | 18.68 | 35.80 | 100.47 |

Table 5: $\mathcal{L}_2$-UVP for our method, NOTB, EgBary ($\epsilon = 0.01$), WIN and FCWB, $D = 2, 4, 8, 16, 64$.

The Table shows that for $\tau = 10^2$, our approach gives the results comparable with the current SOTA solver for continuous **balanced** barycenter estimation $-$ NOTB. Further increase of $\tau$ does not help to improve the results which can be explained by related numerical instabilities. At the same time, discrete OT barycenter solver provides the worst results and the $\mathcal{L}_2$-UVP metric increases drastically with the increase of dimension $D$. It is an expected behaviour, since the discrete distributions poorly approximate the continuous ones. This aspect was previously investigated and justified in (Korotin et al., 2022a, §5.1).

## D.2 SEMI-UNBALANCED OT BARYCENTERS FOR GAUSSIAN DISTRIBUTIONS

A recent preprint (Nguyen et al., 2024) provides an iteration procedure for calculating unbalanced SUOT barycenter for Gaussian distributions using the quadratic cost and KL divergences. **We test our solver in this unbalanced setup for different parameters $\tau$ and show that it consistently outperforms the SOTA balanced solver (Kolesov et al., 2024a, NOTB).**

| Method | $\tau = 10^{-2}$ | $\tau = 10^{-1}$ | $\tau = 1$ | $\tau = 10$ | $\tau = 10^2$ | $\tau = 10^3$ | $\tau = 10^4$ |
|---|---|---|---|---|---|---|---|
| **Ours** | **0.31** | **0.20** | **0.15** | **0.17** | **0.17** | **0.17** | **0.17** |
| NOTB | 1.71 | 1.22 | 0.47 | 0.21 | 0.20 | 0.19 | 0.19 |

Table 6: $\mathrm{B\mathbb{W}}_2^2$-UVP between the learned and ground-truth *unbalanced* barycenters for our method and NOTB. The results are given for varying parameters $\tau = 10^{-2}, 10^{-1}, 1, 10, 10^2, 10^3, 10^4$.

We consider the same set of Gaussians distributions, weights and quadratic cost function as in the experiment with balanced Gaussians, see Appendix D.1. As the baseline solver, we consider SOTA balanced solver for continuous barycenter estimation (Kolesov et al., 2024a, NOTB). To calculate the ground-truth barycenter, we run the *Hybrid Bures-Wasserstein Gradient Descent* algorithm (Nguyen et al., 2024, §4.3) for computation of SUOT barycenter problem between Gaussians. The algorithm is designed for the case of KL divergencies and is endowed with the unbalancedness parameter which we denote as $\tau$.

We consider parameters $\tau \in [10^{-2}, 10^{-1}, 1, 10, 10^2, 10^3, 10^4]$. For each parameter, we calculate the corresponding ground-truth SUOT barycenter and train our U-NOTB solver with KL divergencies. Then we assess the performance of our solver and NOTB by measuring the weighted Bures-Wasserstein unexplained variance percentage metric for an implicitly given barycenters $\widehat{\mathbb{Q}}$:

$$\mathrm{B\mathbb{W}}_2^2\text{-UVP}(\widehat{\mathbb{Q}}) \stackrel{\mathrm{def}}{=} 100 \cdot \Big[ \frac{\mathrm{B\mathbb{W}}_2^2(\widehat{\mathbb{Q}}, \mathbb{Q}^*)}{\frac{1}{2}\mathrm{var}(\mathbb{Q}^*)} \Big]\%$$

where $\mathrm{B\mathbb{W}}_2^2(\mathbb{Q}_1, \mathbb{Q}_2) = \mathbb{W}_2^2(\mathcal{N}(\mu_{\mathbb{Q}_1}, \Sigma_{\mathbb{Q}_1}), \mathcal{N}(\mu_{\mathbb{Q}_2}, \Sigma_{\mathbb{Q}_2}))$ is the Bures-Wasserstein metric, $\mu_{\mathbb{Q}}, \Sigma_{\mathbb{Q}}$ denote means and covariances of corresponding distribution $\mathbb{Q}$, see (Korotin et al., 2021c). To get the samples from learned barycenter $\widehat{\mathbb{Q}}$, we first sample the input points $x_k$ either form the input distributions $\mathbb{P}_k$ (for NOTB) or from the left marginals of the learned plans (for our U-NOTB). Here sampling from the left marginals of the plans can be done using the rejection sampling procedure, see §4.2. Then we sample new points from the learned barycenter by passing the points $x_k$ through the learned stochastic or deterministic maps $T_k$ which parametrize the learned plans. Table 6 presents the weighted sum of $\mathrm{B\mathbb{W}}_2^2$-UVP values w.r.t. the barycenter weights $\lambda_k$.

We see that our unbalanced U-NOTB solver **outperforms** NOTB for **all** considered unbalancedness parameters $\tau$. The difference is especially visible for small parameter $\tau$. It is expected since for small $\tau$ the solutions of the unbalanced barycenter problem significantly differ from the balanced one. For large $\tau$ this difference is getting smaller, thus, balanced NOTB solver manages to approximate the barycenter quite well but still worse than ours.

## D.3 MANIPULATING IMAGES THROUGH INTERPOLATING IMAGE DISTRIBUTIONS ON THE IMAGE MANIFOLD

In this section, we present a novel real-world application of U-NOTB for high-dimensional image manipulation. Specifically, we address scenarios characterized by class imbalances in the marginal distributions. **Ultimately, our experiment highlights the potential of our approach to manipulate images by learning barycenters between distinct image distributions, enabling controlled transitions across semantic attributes. Moreover, our model also demonstrates robustness to class imbalancedness in this practical task compared to other baselines.**

Given multiple image distributions $\{\mathbb{P}_k\}_{k=1}^K$, we aim to simultaneously learn all UOT barycenters $\mathbb{Q}_{\lambda_{1:K}}$ for arbitrary tuple $(\lambda_k)_{k=1}^K$ satisfying $\sum_{k=1}^K \lambda_k = 1$, $0 \le \lambda_k \le 1$. Specifically, we learn all intermediate barycenters simultaneously by parametrizing the transport maps $\{T_k\}_{k=1}^K$ as functions of the condition variable $\lambda_{1:K}$, expressed as $T_k(\lambda_{1:K}, \cdot)$. Note that a similar conditioning on barycenter weights is also employed in Algorithm 2 of (Fan et al., 2021).

With the learned conditional transport plan, we perform image manipulation through transforming given image $x \sim \mathbb{P}_k$ to the barycenter point $y \sim \mathbb{Q}_{\lambda_{1:K}}$. In particular, we consider $\mathbb{P}_1$ and $\mathbb{P}_2$ as collections of images of young individuals (age between 5 to 20) and elderly individuals (age of over 50), respectively, from the FFHQ (Karras et al., 2019) dataset. **It is important to note that the ratios of females to males in $\mathbb{P}_1$ and $\mathbb{P}_2$ are approximately 1:2 and 3:1, respectively. Thus, the data inherently exhibits class imbalance across the marginals.** Here, we provide the precise experimental settings:

- **Marginal distributions $\mathbb{P}_1, \mathbb{P}_2$.** The marginal distributions $\mathbb{P}_1, \mathbb{P}_2$ consist of FFHQ images of young individuals ($5 <$ age $< 20$) and elderly individuals (age $> 50$). We randomly partition each distribution into 90% for the training set and 10% for the test set.

- **StyleGAN Manifold.** Let $G : \mathcal{W}^+ \rightarrow \mathbb{R}^{3 \times 256 \times 256}$ be the pretrained StyleGAN2-ada (Karras et al., 2020) generator that maps $\mathcal{W}^+$ space to 256-dimensional images. Let $E : \mathbb{R}^{3 \times 256 \times 256} \rightarrow \mathcal{W}^+$ be the pretrained FFHQ e4e encoder (Tov et al., 2021). Note that $E$ maps image $x \in \mathbb{R}^{3 \times 256 \times 256}$ to the corresponding vector $w \in \mathcal{W}^+$, i.e. $G(w) \approx x$. Note that $\mathcal{W}^+ \subset \mathbb{R}^{18 \times 512}$. To restrict our transformed images to FFHQ images, we parametrize our transport map by follows: $T_{k,\omega}(x) := G \circ S_{k,\omega} \circ E(x)$ where $x \in \mathbb{R}^{3 \times 256 \times 256}$ and $S_{k,\omega} : \mathbb{R}^{18 \times 512} \rightarrow \mathbb{R}^{18 \times 512}$.

- **General Transport Cost.** To guide image manipulation in a meaningful direction, we adopt a quadratic cost function in the $\mathcal{W}^+$ latent space, defined as:

$$c(x, y) = \alpha \|E(y) - E(x)\|^2. \tag{53}$$

Here, we set $\alpha = 10^{-2}$.

- **Network Parametrization.** Let $(\lambda_1, \lambda_2) := (1 - t, t)$. To generate $\mathbb{Q}_{\lambda_{1:2}}$ from $x \sim \mathbb{P}_k$, we condition the transport map $T_{k,\omega}$ with $t$ as follows:

$$T_{k,\omega}(t, x) = G\left(S_{k,\omega}(t, E(x))\right). \tag{54}$$

We parametrize $S_{1,\omega}(t, z) = z + t\mathrm{NN}_{1,\omega}(t, z)$ and $S_{2,\omega}(t, z) = z + (1-t)\mathrm{NN}_{2,\omega}(t, z)$. Note that $S_{1,\omega}$ and $S_{2,\omega}$ transport the latent vector of young individuals to barycenter point, latent vector of elderly individuals to barycenter, respectively. Thus, we selected this heuristic parametrization of $S_{k,\omega}$ to ensure that $S_{1,\omega}(t, x) \approx x$ when $t \approx 0$ and $S_{2,\omega}(t, y) \approx y$ when $t \approx 1$. Furthermore, the potential functions $f_1, f_2$ are also conditioned by variable $t$ as follows:

$$f_{1,\theta}(t, x) = \frac{V_\theta(t, E(x))}{1 - t} + m_\theta(t), \ f_{2,\theta}(t, x) = -\frac{V_\theta(t, E(x))}{t} + m_\theta(t), \tag{55}$$

where $V_\theta : \mathbb{R} \times \mathbb{R}^{18 \times 512} \rightarrow \mathbb{R}$. Note that $(1 - t)f_{1,\theta} + tf_{2,\theta} = m_\theta(t)$. Here, $m_\theta(t)$ can be regarded as the parametrization of the congruence constant at time $t$. To avoid the numerical instability of (55) with respect to the time variable $t$, we sample $t$ from a uniform distribution within the interval $[0.05, 0.95]$.

**Results.** As discussed, the two marginals $\mathbb{P}_1, \mathbb{P}_2$ exhibit a class imbalance with respect to gender. $\mathbb{P}_1$ contains more than 60% of female individuals, while $\mathbb{P}_2$ consists of less than 30% of female individuals. Thus, we evaluate the quantitative performance of our transport map by measuring how well the transport map preserves gender in such a class imbalanced case. Specifically, we measure the gender preservation accuracy when transforming a young individual $x$ to barycenter $T_1(t, x)$ at $t = 0.9$. For our model, we report the gender preservation accuracy for the accepted images. As a comparison, we include the results from the NOTB method applied to the entire test dataset. For the baseline model, we precompute the global latent direction $v_{global} := \mathbb{E}_{y \sim \mathbb{P}_2}[E(y)] - \mathbb{E}_{x \sim \mathbb{P}_1}[E(x)]$, and measure the accuracy of the gender alignment between $x \sim \mathbb{P}_1$ and $x + tv_{global}$.

| Method | Baseline | NOTB | U-NOTB |
|---|---|---|---|
| Acceptance Rate | - | - | 67.9% (592 / 872) |
| Accuracy | 69.8% (609 / 872) | 63.4% (553 / 872) | **88.0%** (521 / 592) |

Table 7: Gender preservation accuracy for $x \sim \mathbb{P}_1$ and $T_1(t, x)$ at $t = 0.9$. U-NOTB can perform rejection sampling to address class imbalancedness.

As shown in Table 7, our model achieves a significantly high accuracy of 88%, highly outperforming other methods, which achieve less than 70%. This demonstrates that our model performs robustly

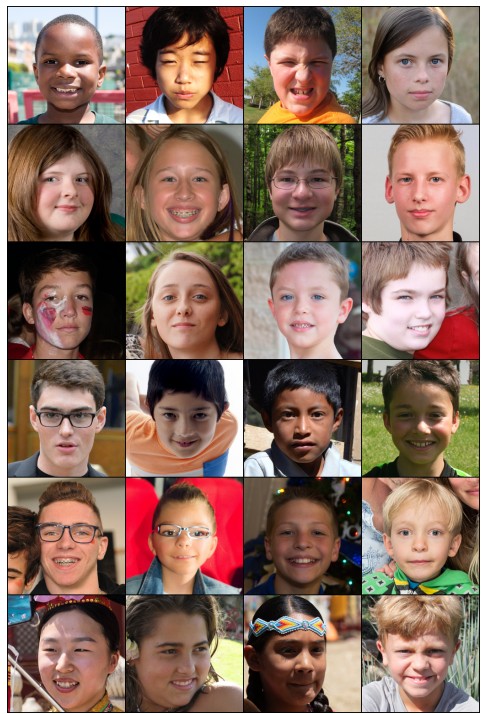
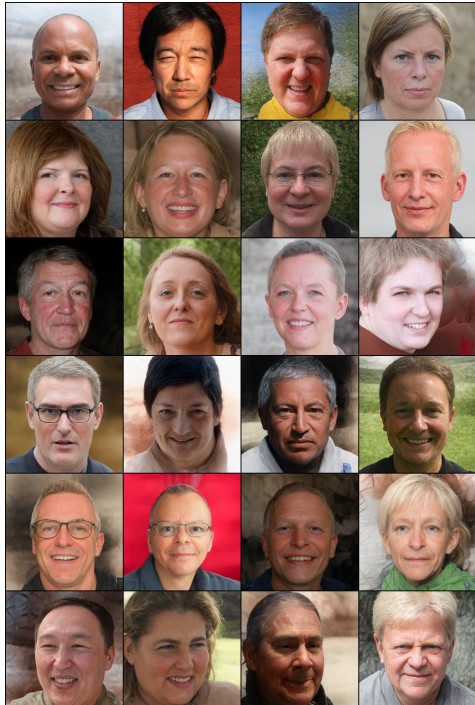

(a) Accepted samples from $\mathbb{P}_1$.

(b) Corresponding barycenter samples.

Figure 6: Examples of randomly selected (a) accepted samples of $x \sim \mathbb{P}_1$ and (b) its corresponding barycenter at $t = 0.9$. Note that the acceptance rate of U-NOTB is 67.9%. Approximately, 36% of the accepted samples are female individuals.

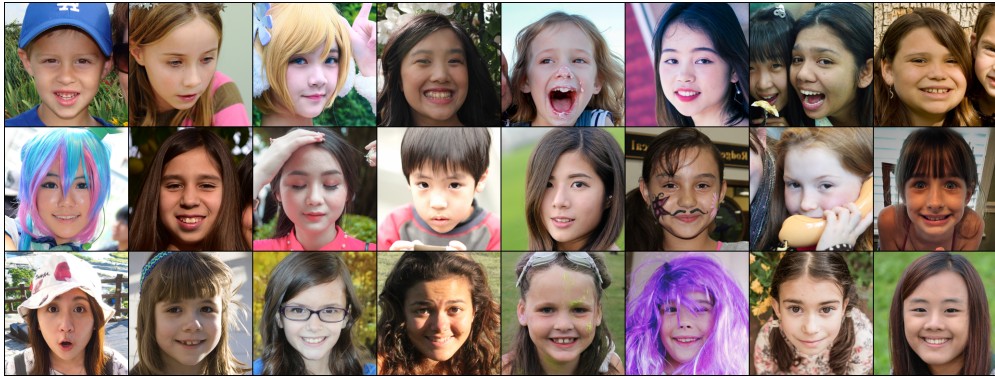

Figure 7: Examples of rejected samples of U-NOTB. Approximately 80% of rejected samples are female images.

under class-imbalanced conditions. Moreover, only 36.1% of the accepted samples were female. Given that over 60% of the data in $\mathbb{P}_1$ consisted of female samples, this result indicates that a significant proportion of female samples were rejected to address the class imbalance. For qualitative examples, please refer to Figure 6 and Figure 7.

**Evaluation Metric.** To classify the gender of images, we use the open python library called OpenCV. Specifically, we adjusted the code from the following github address:

```
https://github.com/smahesh29/Gender-and-Age-Detection
```

**Implementation Details.** We set $(\bar{\psi}_1, \bar{\psi}_2) = (\text{Softplus}, \text{Softplus})$ and $\tau = 0.1$. We employ the learning rate of $10^{-5}$, batch size of 4, and the total number of iterations of 5K. For all the networks $\text{NN}_{1,\omega}, \text{NN}_{1,\omega}, V_\theta$ we employ the same embedding approach: the time variable $t$ is embedded

| D | $512 \times 18$ |
|---|---|
| K | 2 |
| $\tau$ | 0.1 |
| batch size | 4 |
| # Iterations | 5000 |
| $N_T$ | 1 |
| $\lambda_1$ | 1/2 |
| $\lambda_2$ | 1/2 |
| Divergence $\psi_1$ | Softplus |
| Divergence $\psi_2$ | Softplus |
| $f_{k,\theta}$ | Transformer |
| $T_{k,\omega}$ | Transformer |
| $lr_{f_{k,\theta}}$ | 1e-5 |
| $lr_{T_{k,\omega}}$ | 1e-5 |
| $lr_m$ | 1e-5 |
| Training Time | 4-5 h |
| GPU | RTX 3090Ti |

Table 8: Hyper-parameter settings and training time for FFHQ experiment.

into 512-dimension vector $t_{emb} \in \mathbb{R}^{1 \times 512}$ through sinusoidal embedding, and concatenated with $z \in \mathbb{R}^{18 \times 512}$. For $\text{NN}_{1,\omega}, \text{NN}_{2,\omega}$, these embeddings are processed through two transformer layers, each with four attention heads. Then, we pass through the linear layer with both input and output channels of 512, resulting the output of size $\mathbb{R}^{18 \times 512}$. For the potential $V_\theta$, we use pass through four transformer layers, each with four attention heads. Then, we aggregate features by taking mean of the tokens, which is then passed through the final linear layer. Additionally, for the network $m_\theta(t)$, we embed $t$ by sinusoidal embedding, and then pass it through 2-layered MLP. The outline of the hyperparameter settings and the training time are described in Table 8.

## E    LIMITATIONS AND FUTURE RESEARCH DIRECTIONS

In practice, we found that the practical optimization procedure (Algorithm 1) may work unstably. In particular, the training may diverge under improper hyper-parameters selection or the resulting quality might depend on the random seed. We hypothesise that this behaviour is due to the reliance on adversarial training and leave its thorough investigation to future research. From the theoretical side, the recovered semi-unbalanced OT barycenter is not guaranteed to be unique (§2.2). Resolving these limitations paves a way for future work. Another appealing direction for future research is to adopt alternative robust OT/barycenter variants, e.g., (Nietert et al., 2022; Buze, 2024), for the continuous barycenter setup.

