# OpenReview forum: "Robust Barycenter Estimation using Semi-Unbalanced Neural Optimal Transport"
_ICLR.cc/2025/Conference — ICLR 2025 Poster_

### Official Review · Reviewer_niQu · 2024-10-27

**Soundness:** 4
**Presentation:** 4
**Contribution:** 3
**Rating:** 8
**Confidence:** 4

**Summary:**

In this work, authors propose to solve the continuous unbalanced optimal transport barycenter. To do that, they derive a dual formulation to the problem, which is a min-max problem over potentials and conditional distributions. They solve the problem by parametrizing the potential and conditional distributions with neural networks. Finally, they show that their method works on three experiments, demonstrating that they recover the right barycenters, that the barycenter is robust to outliers and to class imbalance, and that it can handle using different costs.

**Strengths:**

The paper is overall well written and nice to read.

It provides one of the first method to solve continuous UOT barycenters leveraging a dual formulation and neural networks parametrizations, which is an interesting contribution.

Several convincing experiments are done demonstrating that the method learns well the UOT barycenter, that it is robust to outliers and class imbalance, and that it works with several costs.

**Weaknesses:**

The main weakness in my opinion is that the method feels really incremental compared to [1]. The main difference is that it it adapted to the UOT problem, which just changes slightly the formulation of the dual, and how to sample from the barycenter for inference.

Another weakness, which is classical with UOT, is that the choice of the unbalancedness parameters does not seem easy.

The method also needs to solve a min-max problem, which is probably unstable and very costly.

**Questions:**

A term seems to be missing in the definition of $\psi$-divergence, see Definition 1 in [2].

The sentence line 227 is not clear to me. The $m$-congruence is a constraint of the problem. So I do not understand why it is written "if the potentials satisfy the m-congruence, then the optimal value of the SUOT barycenter can be derived by solving (8)". The SUOT barycenter can always be derived by solving (8) and the potentials necessarily satisfy this constraint when solving (8).

In Corollary 1, it is stated that the sup is taken over $f_{[1,K]}$. Isn't it also taken over $m$?

In Theorem 2, equation (12), shouldn't it be an argmin?

In Section 5.1, it is stated that the UOT problem is equivalent with the OT problem between rescaled distribution. Is it truly equivalent? And could we solve the OT barycenter between the rescaled distributions instead of solving the SUOT barycenter?

Is it proved that $T_{1\\#}\mathbb{P}_1$ gives the UOT barycenter?

I think that the reference [3] is missing. In their experiments, they compute robust OT barycenters via unbalancedness.

Typos:
- Line 11: The first sentence of the abstract feels weird: I don't see what is the "common challenge"
- Line 107: "This transition is valid since we work the infimum in weak"
- Line 142: "wights"
- Line 365: what is $T$ in $T_1 = \lambda_1 Id + \lambda_2 T$
- Legend of Figure 4: $\mathbb{P}_0$ -> $\mathbb{P}_1$ and $\mathbb{P}_1$ -> $\mathbb{P}_2$

[1] Kolesov, A., Mokrov, P., Udovichenko, I., Gazdieva, M., Pammer, G., Burnaev, E., & Korotin, A. (2024). Estimating Barycenters of Distributions with Neural Optimal Transport. arXiv preprint arXiv:2402.03828.

[2] Séjourné, T., Peyré, G., & Vialard, F. X. (2023). Unbalanced optimal transport, from theory to numerics. Handbook of Numerical Analysis, 24, 407-471.

[3] Séjourné, T., Bonet, C., Fatras, K., Nadjahi, K., & Courty, N. (2023). Unbalanced optimal transport meets sliced-Wasserstein. arXiv preprint arXiv:2306.07176.

---

> ### Author Response · Authors · 2024-11-22
> **Response to Reviewer niQu**
>
> Thank you for your detailed feedback. Please find the answers to your questions below.
>
> **(1) The main weakness in my opinion is that the method feels really incremental compared to (Kolesov et al., 2024, Estimating). The main difference is that it is adapted to the UOT problem, which just changes slightly the formulation of the dual, and how to sample from the barycenter for inference.**
>
> We would like to clarify the novelty of our paper.
> It would be most correct to position our method as a generalization of the recent SOTA approach to estimating continuous **balanced** OT barycenters - **NOTB** (Kolesov et al., 2024, Estimating), to the case of **semi-unbalanced OT** barycenters. However, this generalization is not straightforward as it is based on different principles.
>
> *First*, the congruence condition on the potentials, which is used in the NOTB approach, ceases to be true in the case of semi-unbalanced OT. In our paper, we developed a completely different condition on the potentials, which required the usage of an additional parameter $m$ in the optimization algorithm.
>
> *Second*, the procedure of sampling from the barycenter during the inference has major differences from the one used in NOTB. In the semi-unbalanced case, to get samples from the barycenter, we first need to define the procedure of sampling from the left marginals of the plans. Defining such a sampling procedure is tricky even for the continuous unbalanced OT (UOT) solver by its own. As far as we know, only one of the existing continuous UOT solvers (Gazdieva et al., 2024) propose such a strategy using the **restricted** Gaussian mixture parametrization of potentials. Other scalable and theoretically justified continuous UOT solvers (Choi et al., 2024, Yang et al., 2018) do not offer such a precise sampling procedure and consider only sampling from the input measure. In contrast to previous approaches, we suggest a precise sampling procedure that utilizes equation (11) from our paper.
> **Using the learned potentials, our solver allows for precise sampling of barycenter from the left marginals of learned plans without any restrictions on the underlying potentials.**
> In the case of previous works in the field such as NOTB, the situation is much easier since these marginals coincide with input measures and sampling is defined a priori.
>
> *Third*, during the rebuttal period, we conducted **a novel high-dimensional experiment** to further demonstrate the practical advantages of our method. Specifically, as detailed in Appendix C.3, we performed interpolation between two image distributions within the FFHQ 256×256 dataset, focusing on transitioning from the *young* to *elderly* categories. This experiment highlights the potential of our approach to manipulate images by learning barycenters between distinct image distributions, enabling controlled transitions across semantic attributes. Moreover, our model also demonstrates robustness to class imbalancedness in this practical task compared to other baselines.
>
> *More detailed explanation regarding this experiment is given in Appendix C.3 in the revised version of our paper.*
>
> **(2) Another weakness, which is classical with UOT, is that the choice of the unbalancedness parameters does not seem easy.**
>
> Indeed, the choice of unbalancedness parameter is a tricky point of *any method* related to UOT problem. In our paper, we followed recent works in Unbalanced Optimal Transport (UOT), such as (J. Choi et al., 2023, J. Choi et al., 2024), and adopted a manual selection of the unbalancedness parameter $\tau$. Still, we believe that this flexibility in choosing $\tau$ can be interpreted as an advantage because $\tau$ can be tailored to suit the specific properties of the distributions, e.g., control the amount of samples treated as outliers. However, we also believe that developing a method to automatically determine $\tau$ based on the proportion of outliers or class imbalance would be a highly interesting direction for future research.

---

> > ### Author Response · Authors · 2024-11-22
> > **Response to Reviewer niQu (continuation)**
> >
> > **(3) The method also needs to solve a min-max problem, which is probably unstable and very costly.**
> >
> > We agree that the reliance on adversarial training (with all its pitfalls) is indeed a limitation of our method (we mention this in Section 6). However, we want to underline the methodological/computational complexity of the problem we solve in our paper (OT barycenter $-$ and even its more tricky *unbalanced* version). In fact, this complexity is proclaimed by the non-trivial (and far from being easily implemented and made to work) methodological choices of other existing barycenter solvers. In particular, (Kolesov et. al., 2024, Estimating) and (Korotin et. al., 2022) rely on adversarial training similar to us; (Kolesov et. al., 2024, Energy) use iterative Langevin sampling at the training and inference stages which is costly. To the best of our knowledge, the earlier OT barycenter approaches (see our Related works - Section 3) either also rely on some rather complex techniques, or consider too narrow OT barycenter formulations (e.g., Wasserstein-2 barycenters). Note that the latter makes applications similar to Color-Shape experiment from our paper (Section 5.3) impossible.
> >
> > **(4) The sentence line 227 is not clear to me. <...>**
> >
> > Thanks for pointing to this sentence, we updated it in the revised version of our paper.
> >
> > **(5) A term seems to be missing in the definition of $\psi$-divergence, see Definition 1 in (Séjourné et al., 2023b).**
> >
> > The "term" which you are referring to disappears in the case when $\psi$-divergence is defined between the measures $\mu\_1$, $\mu\_2$ s.t. $\mu\_1\ll\mu\_2$. In our paper, we deal with measures which satisfy this property $-$ thus, we decided to not overload the text by introducing this "term". However, to make this aspect more rigorous, we fix the definition of $\psi$-divergence in the revised version of our paper by specifying that ${\mathcal{D}\_{\psi}(\mu\_1\|\mu\_2)= \int_{\mathcal{X}} \psi\bigg(\frac{\mu\_{1}(x)}{\mu\_{2}(x)}\bigg)d\mu\_{2}(x)}$ only if $\mu\_{1}\ll \mu\_2$ and $+\infty$ otherwise.
> >
> > **(6) In Corollary 1, it is stated that the sup is taken over $f_k$. Isn't it also taken over $m$?**
> >
> > Yes, it is written in the Eq. (9) of Corollary 1 but accidentally is not included in the corresponding caption. Thanks for noting, we added it in the revised version of our paper.
> >
> > **(7) In Theorem 2, equation (12), shouldn't it be an argmin?**
> >
> > Since the infimum in equation (12) is attained at least for one map $\gamma\_k^*(\cdot|x\_k)$, the $\arg\inf\_{\gamma\_k(\cdot|x\_k))}$ can be certainly replaced by $\arg\min_{\gamma\_k(\cdot|x\_k))}$. We updated Equation 12 accordingly.
> >
> > **(8) In Section 5.1, it is stated that the UOT problem is equivalent with the OT problem between rescaled distribution. Is it truly equivalent? And could we solve the OT barycenter between the rescaled distributions instead of solving the SUOT barycenter?**
> >
> > The equivalence between the solutions of the UOT problem and OT problem between the rescaled marginals is shown in (Choi et al., 2024), see their Theorem 3.3. We also include this theorem in our Appendix A.2 since we used it in the proof of our Theorem 2. Specifically, we exploited in that proof the *equivalence* between the unbalanced OT barycenter problem and its balanced counterpart. To ease the explanations, we recall this connection below.
> >
> > In principle, we can reformulate the semi-unbalanced OT barycenter problem as a balanced OT barycenter one:
> > $$
> > \inf\_{\mathbb{Q}\in\mathcal{P}(\mathcal{Y})} \mathcal{B}\_{u}(\mathbb{Q})=\inf_{\mathbb{Q}\in\mathcal{P}(\mathcal{Y})}\sum\_{k=1}^K \lambda\_k \text{SUOT}\_{c_k,\psi_k}(\mathbb{P}\_k, \mathbb{Q})=\inf\_{\mathbb{Q}\in\mathcal{P}(\mathcal{Y})}\sum_{k=1}^K \lambda_k \text{OT}(\widetilde{\mathbb{P}}\_k, \mathbb{Q})=
> > $$
> > $$
> > \inf\_{\mathbb{Q}\in\mathcal{P}(\mathcal{Y})}\sum\_{k=1}^K \lambda_k\Bigg\lbrace\sup\_{f_k} \int\_{\mathcal{X}\_k} f_k^c(x\_k) d \widetilde{\mathbb{P}}\_k(x\_k) + \int\_{\mathcal{Y}}  f\_k(y) d \mathbb{Q}(y)\Bigg\rbrace.
> > $$
> > Here the distributions $\widetilde{\mathbb{P}}\_k$ ($k\in\overline{K}$) are specified via the optimal potential $f^*\_k$ delivering maximum to the inner $\sup$ problem: $d\widetilde{\mathbb{P}}_k(x_k)=\nabla \overline{\psi}(-(f^*_k)^c(x_k))d\mathbb{P}_k(x_k)$. This is the main **cornerstone of the balanced reformulation** $-$ prior to solving the balanced OT barycenter problem, we need to identify the re-scaled input measures, i.e., compute the re-scaling factors which can be identified only when the solutions of this problem (optimal potentials) are already given. *Thus, solving this balanced OT barycenter problem between the re-scaled measures seems to be an ambiguous task.*

---

> ### Author Response · Authors · 2024-11-22
> **Response to Reviewer niQu (continuation #2)**
>
> **(9) Is it proved that $T\_1\sharp\mathbb{P}\_1$ gives the UOT barycenter?**
>
> As shown in Theorem 2, the optimal plan $T^\star\_1$ lies in the saddle point solution of our max-min optimization problem (equation 12). Moreover, as shown in equation 11, $\\mathbb{Q} = T^{\\star}_{1}\\# \widetilde{\\mathbb{P}}_1$, where $\widetilde{\mathbb{P}}\_1 = \nabla \bar{\psi} (-(f^\star\_1)^c (x) ) \mathbb{P}\_1$.
> Thus, to obtain a point of the barycenter, we first sample $x\sim \widetilde{\mathbb{P}}$ by rejection sampling (line 327-338). Then, we pass $x$ through our learned transport map $T_1$. *Could you please reply, does this answer your question?*
>
> **(10) I think that the reference (Séjourné et al., 2023a) is missing. In their experiments, they compute robust OT barycenters via unbalancedness.**
>
> Thank you for suggesting the relevant paper. We added the reference in Related Work section of the revised paper. Based on our understanding of the paper, we attribute it as the discrete method.
>
> **(11) Typos.**
>
>
> Thank you for your carefullness, we have fixed all the typos which you have mentioned.
>
>
> **Concluding remarks**. Please respond to our post to let us know if the clarifications above suitably address your concerns about our work. We are happy to address any remaining points during the discussion phase; if the responses above are sufficient, we kindly ask that you consider raising your score.
>
> **References.**
>
> J. Choi, J. Choi, and M. Kang. Generative modeling through the semi-dual formulation of unbalanced optimal transport. In Advances in Neural Information Processing Systems, volume 36, 2024.
>
> M. Gazdieva, A. Asadulaev, E. Burnaev, and A. Korotin. Light Unbalanced Optimal Transport. In Advances in Neural Information Processing Systems, volume 36, 2024.
>
> Kolesov et. al., Estimating Barycenters of Distributions with Neural Optimal Transport, Proceedings of the 41st International Conference on Machine Learning, PMLR 235:25016-25041, 2024.
>
> Séjourné, T., Bonet, C., Fatras, K., Nadjahi, K., \& Courty, N. (2023a). Unbalanced optimal transport meets sliced-Wasserstein. arXiv preprint arXiv:2306.07176.
>
> Séjourné, T., Peyré, G., \& Vialard, F. X. (2023b). Unbalanced optimal transport, from theory to numerics. Handbook of Numerical Analysis, 24, 407-471.
>
> K. D. Yang and C. Uhler. Scalable unbalanced optimal transport using generative adversarial networks. In International Conference on Learning Representations, 2018.
>
> Korotin et al. Wasserstein iterative networks for barycenter estimation. Advances in Neural Information Processing Systems 35, 15672-15686, 2022.
>
> Kolesov et. al., Energy-Guided Continuous Entropic Barycenter Estimation for General Costs, NeurIPS, 2024.

---

> > ### Comment · Reviewer_niQu · 2024-11-23
> >
> > Thank you for your answer and for revising the paper.
> >
> > I have a last question about the method, and also the equivalence between OT and UOT barycenters. Do I understand correcty that the method is only developed to work to handle probability distributions, and would not work with arbitrary positive measures? The same questions holds for the equivalence between OT and UOT barycenters?
> >
> > Otherwise, my concerns and the main weaknesses have been addressed. In particular, the originality of the paper has been clarified. I will update my score to 8.

---

> ### Author Response · Authors · 2024-11-23
> **Response to Reviewer niQu**
>
> Yes, the method is specifically developed for probability distributions. We believe that extending our method to arbitrary positive distributions represents a promising direction for future work.
>
> Moreover, to establish the connection between OT and SUOT barycenters in the case of probability measures we employ the connection between OT and UOT problems established in (Choi et al., 2024). Since the latter result holds true for the case of probability measures, the connection between OT and SUOT barycenters in the case of arbitrary positive measures also needs further investigations.
>
> We are delighted to hear that your concerns have been addressed. Once again, we sincerely appreciate the time and effort you have put into reviewing our paper.

---

### Official Review · Reviewer_hyxj · 2024-10-31

**Soundness:** 3
**Presentation:** 3
**Contribution:** 2
**Rating:** 5
**Confidence:** 4

**Summary:**

In this paper, the authors investigate the problem of finding the barycenter with semi-unbalance optimal cost (SUOT) from perspective of dual theory. In particular:

(1) They derive the theory for the dual form of SUOT barycenter with reformulation, and necessary condition based on marginal plan and conditional plan

(2) Then, from the necessary condition,  they propose the an algorithm (Algorithm 1) to approximate the optimal transport plan from the solution. This algorithm is based on training the neural network of optimal plan.

(3) Then, they empirically validate the efficacy of the algorithm on both synthetic and real dataset.

**Strengths:**

1. The paper provides an important and novel dual theory for the SUOT barycenter problem.

2. The proposed algorthm works well and persistent to the imbalance and outliers.

3. The paper is well-structured and easy to follow.

**Weaknesses:**

1. The necessary condition seems to be not difficult to obtain, which makes it not convincing enough. So, would it be possible to obtain some sufficient conditions?

2. It would be better if the authors can highlight the advantages of finding the optimal plan over the optimal solution.

3. As stated in the limitations, the authors did not provide a rigorous analysis for the convergence of the algorithm.

**Questions:**

See Weaknesses section.

---

> ### Author Response · Authors · 2024-11-22
> **Response to Reviewer hyxj**
>
> Thank you for your detailed feedback. Please find the answers to your questions below.
>
> **(1) The necessary condition seems to be not difficult to obtain, which makes it not convincing enough. So, would it be possible to obtain some sufficient conditions?**
>
> We understood the term "sufficient condition" as follows:
> for every optimal saddle point $\{(f\_k^*, \gamma\_k^*)\}\_{k=1}^K$ of our optimization objective (9), it holds that $\{\gamma\_k^*\}\_{k=1}^K$ is the family of true SUOT plans between $\mathbb{P}\_{[1:K]}$ and $\mathbb{Q}$.
>
> Actually, the latter seems to be true under some additional assumptions related to the *strong convexity* of $c_k(x_k,y)-f_{k}(y)$, ($k\in[1,K]$). These assumptions were already established and studied in the previous works on balanced OT, see (Fan et al., 2021) or (Makkuva et al., 2020), and balanced OT barycenter problem, see (Kolesov et al., 2024, Estimating). In principle, we think that one can make the same kind of assumptions and presumably derive analogous bounds for our semi-unbalanced barycenter setup. However, these assumptions are far from practice since they usually require the usage of restricted class of neural network architectures, i,e., Input Convex Neural Networks (ICNNs). At the same time, the methods exploiting this type of networks are known to provide quite fair results in the tasks of generative modeling, see, e.g., Fig. 4 of (Korotin et al., 2021) for visualization of their performance.
>
> Another way for ensuring the "sufficient condition" consists in considering more general OT formulations, e.g., using different regularizations such as entropic (Gushchin et al., 2024) or kernel (Korotin et al., 2023). We believe that our ideas for basic OT presented in the paper can be generalized to such cases, but leave the investigation of this aspect for future work.
>
> **(2) It would be better if the authors can highlight the advantages of finding the optimal plan over the optimal solution.**
>
> We interpreted this question as asking: Why do we focus on finding an optimal plan instead of directly computing the barycenter distribution? What are the practical advantages of this approach?
>
> There exist a lot of practical tasks which require finding a shared representation of the data which comes from different sources. This task can be positioned as the problem of finding the barycenter of distributions. Then, in order to translate new data from each of the input distributions to the shared representation (barycenter), the practitioner needs to have access to the corresponding translation maps (conditional plans). It explains the importance of finding the optimal plans and not only barycenters by their own. More details on the specific applications where such kind of tasks appear, e.g., finding shared representatons for scans from different MRI scanners or mixing geological simulators, can be found in (Appendix B.2, Kolesov et al., 2024).
>
>
> **(3) As stated in the limitations, the authors did not provide a rigorous analysis for the convergence of the algorithm.**
>
> Deriving theoretical results regarding the convergence of the obtained algorithm for estimating continuous unbalanced OT barycenters is a difficult and even might be not a solvable task under the reasonable conditions. Indeed, even existing results of this kind for the UOT problem itself seem to require restrictive assumptions which are not feasible in practice. For example, (Choi et al., 2023) in their Theorem 3.4 assume strong convexity of the potentials, which is not applicable in practice, as we explained in the first answer to you.
>
> **Concluding remarks**. Please respond to our post to let us know if the clarifications above suitably address your concerns about our work. We are happy to address any remaining points during the discussion phase; if the responses above are sufficient, we kindly ask that you consider raising your score.

---

> > ### Author Response · Authors · 2024-11-22
> > **Response to Reviewer hyxj (references)**
> >
> > **References.**
> >
> > Jiaojiao Fan, Amirhossein Taghvaei, and Yongxin Chen. Scalable computations of wasserstein
> > barycenter via input convex neural networks. In Marina Meila and Tong Zhang (eds.), Proceedings
> > of the 38th International Conference on Machine Learning, volume 139 of Proceedings of Machine
> > Learning Research
> >
> > Korotin, A., Selikhanovych, D., \& Burnaev, E. Kernel Neural Optimal Transport. In The Eleventh International Conference on Learning Representations, 2023
> >
> > Kolesov et. al., Energy-Guided Continuous Entropic Barycenter Estimation for General Costs, NeurIPS, 2024.
> >
> > Makkuva, A., Taghvaei, A., Oh, S., \& Lee, J. (2020, November). Optimal transport mapping via input convex neural networks. In International Conference on Machine Learning (pp. 6672-6681). PMLR.
> >
> > Korotin, A., Li, L., Genevay, A., Solomon, J. M., Filippov, A., \& Burnaev, E. (2021). Do neural optimal transport solvers work? a continuous wasserstein-2 benchmark. Advances in neural information processing systems, 34, 14593-14605.
> >
> > Gushchin, N., Kolesov, A., Korotin, A., Vetrov, D. P., \& Burnaev, E. (2024). Entropic neural optimal transport via diffusion processes. Advances in Neural Information Processing Systems, 36.
> >
> > Choi, J., Choi, J., and Kang, M.. Generative modeling through the semi-dual formulation of unbalanced optimal transport. In Advances in Neural Information Processing Systems, volume 36, 2023.
> >
> > Kolesov et. al., Estimating Barycenters of Distributions with Neural Optimal Transport, Proceedings of the 41st International Conference on Machine Learning, PMLR 235:25016-25041, 2024

---

> > > ### Comment · Reviewer_hyxj · 2024-11-24
> > >
> > > Dear the Authors,
> > >
> > > Thanks for your response. However, I am still not convince by your answer to my concern about the convergence analysis of the algorithm. I think such analysis is very important to investigate the theoretical properties of the optimal solution, and to determine whether that algorithm is practical or not. Given this reason, I decided to keep my rating unchanged.

---

> > > > ### Author Response · Authors · 2024-11-28
> > > > **Additional comment for Reviewer hyxj**
> > > >
> > > > Dear Reviewer,
> > > >
> > > > as per your request, we have included in the revised version of our paper a new theoretical result (Theorem 3 in Section 4) showing the quality bounds for the recovered plans according to the duality gaps, i.e., the errors for solving inner and outer optimization problems in our objective (9). Thanks to this result, we deduce that when our Algorithm 1 optimizing (9) converged nearly to the optimum, its solutions are close to the true conditional plans.
> > > >
> > > > We will appreciate if you could take into account this requested result when finalizing your score.
> > > >
> > > > Best regards,
> > > > the Authors

---

> > > > > ### Author Response · Authors · 2024-12-02
> > > > >
> > > > > Dear reviewer,
> > > > >
> > > > > As the rebuttal phase deadline is approaching in a few hours, we would greatly appreciate your feedback on our responses to the reviews.
> > > > >
> > > > > Thank you.
> > > > >
> > > > > Best regards, The Authors

---

> > > > > > ### Author Response · Authors · 2024-12-03
> > > > > >
> > > > > > Dear reviewer,
> > > > > >
> > > > > > The time of discussion between authors and reviewers is coming to an end. We have made our best to address you concerns about theoretical properties of the solutions of our algorithm. In our new Theorem 3, we establish the conditions under which the plans estimated by our approach are close to the true UEOT ones which justifies the practical usability of our approach. This proof is far from being trivial and we will be happy to address any of your questions.
> > > > > >
> > > > > > We kindly ask you to give us the feedback regarding this new result.

---

### Official Review · Reviewer_MXNH · 2024-11-03

**Soundness:** 3
**Presentation:** 3
**Contribution:** 2
**Rating:** 6
**Confidence:** 4

**Summary:**

This paper proposes a scalable approach for estimating the robust continuous barycenter by using the dual formulation of the (semi-)unbalanced OT problem. They first attempt to develop an algorithm for robust barycenters under continuous distribution setup. They model this problem as a min-max optimization problem and provide theoretical underpinnings, along with experimental results on synthetic and real-world datasets to demonstrate the robustness and adaptability of their method.

**Strengths:**

- This paper introduces a continuous SUOT barycenter estimation method that addresses the issue of robustness against outliers and imbalances in real-world datasets.
- Rigorous derivation of the SUOT-based framework and solid theoretical support for the proposed model give the approach a solid foundation.

**Weaknesses:**

Experiments are not sufficient and the baseline method is not comprehensive.

**Questions:**

- Given that the Wasserstein barycenter of Gaussian distributions has a closed-form solution, could the authors provide experimental verification for the Gaussian distribution case?
- Can the robustness parameter $\tau$ be automatically optimized during training rather than manually tuned? If not, for different proportions or types of noise, different $tau$ is usually required. Will this limit the practicality of the scheme?
- How does the method work with high-dimensional data? In other words, the support size of the measures may be large in practical applications. While the support size of the measures in your experiments seems rather small.
- Why don't you compare the methods [1,2] based on robust OT? Please add corresponding experiments.

[1] Nietert S, Goldfeld Z, Cummings R. Outlier-robust optimal transport: Duality, structure, and statistical analysis[C]//International Conference on Artificial Intelligence and Statistics. PMLR, 2022: 11691-11719.

[2] Wang X, Huang J, Yang Q, et al. On Robust Wasserstein Barycenter: The Model and Algorithm[C]//Proceedings of the 2024 SIAM International Conference on Data Mining (SDM). Society for Industrial and Applied Mathematics, 2024: 235-243.

---

> ### Author Response · Authors · 2024-11-22
> **Response to Reviewer MXNH**
>
> Thank you for your detailed feedback. Please find the answers to your questions below.
>
> **(1) Given that the Wasserstein barycenter of Gaussian distributions has a closed-form solution, could the authors provide experimental verification for the Gaussian distribution case?**
>
> Thank you for this valuable suggestion. In the revised version of our paper, we test our solver in the balanced/semi-unbalanced OT barycenter problem for Gaussian distributions with computable *ground-truth solutions*.
>
> In the balanced OT barycenter problem for Gaussian distributions, the ground-truth barycenter is known to be Gaussian and can be estimated using the fixed point iteration procedure (Álvarez-Esteban et al., 2016). We tested our solver in this balanced setup keeping in mind that for large unbalancedness parameters $\tau$, it should provide a good approximation of balanced OT barycenter problem solutions. **Ultimately, we show that while our solver is designed to tackle an *unbalanced* OT barycenter problem, its performance in the different balanced OT barycenter problem is comparable to the current SOTA solvers.**
>
> A recent preprint (Nguyen et al., 2024)  provides an iteration procedure for calculating unbalanced SUOT barycenter for Gaussian distributions using the quadratic cost and KL divergences. **We test our solver in this *unbalanced* setup for different parameters $\tau$ and show that it consistently outperforms the SOTA balanced solver (Kolesov et al., 2024a).**
>
> *We include these new requested experiments in Appendices C.1, C.2, of the revised version of our paper.*
>
> **(2) Can the robustness parameter  $\tau$ be automatically optimized during training rather than manually tuned? If not, for different proportions or types of noise, different $\tau$ is usually required. Will this limit the practicality of the scheme?**
>
> We appreciate the reviewer for highlighting an important aspect of our approach. Following recent works in Unbalanced Optimal Transport (UOT), such as (J. Choi et al., 2023, J. Choi et al., 2024), our model adopts a manual selection of the robustness parameter $\tau$. Meanwhile, we believe that this flexibility in choosing $\tau$ can be interpreted as an advantage because $\tau$ can be tailored to suit the specific properties of the distributions, e.g., control the amount of samples treated as outliers.  However, we also believe that developing a method to automatically determine $\tau$ based on the proportion of outliers or class imbalance would be a highly interesting direction for future research.
>
> **(3)  How does the method work with high-dimensional data? In other words, the support size of the measures may be large in practical applications. While the support size of the measures in your experiments seems rather small.**
>
> We are not quite sure what do you exactly mean by "*support*" in this question because this notion is usually treated differently by researchers from discrete and continuous OT field. In the case of discrete OT, "*support*" usually corresponds to the size of the dataset. In the case of continuous OT, it is treated as an ambient or intrinsic data dimension. Thus, we provide the answers for both sides of this question.
>
> *Dataset size.* The sizes of the datasets do not matter for our algorithm. It uses the stochastic optimization and can handle arbitrarily large datasets.
>
> *Data space dimension.*  Searching for the barycenters directly in the data space may not be very meaningful because, as a result of averaging, some practically meaningless objects may appear. For example, averaging images of '0' and '1' with $\ell_2$ cost directly in the image space will result in straightforward $\ell_2$ interpolation of these images. Thus, dealing with a barycenter problem,  it is necessary to restrict the space where we search for this barycenter. For example, this space can be specified using the pretrained generative models, e.g., StyleGAN (Karras et al., 2019), as it was done in the previous related papers (Kolesov et al., 2024a,b) and our experiment with MNIST dataset, see our Section 5.3. Here the data space dimension usually means an intrinsic dimension, i.e., the dimension of the manifold where this data lies. In the context of generative models, this intrinsic dimension is usually treated as a dimension of the model's latent space. Our experiments in Section 5.3 are conducted in standard StyleGAN latent space of the dimension 512.
>
> (See the next comment for continuation of this answer.)

---

> > ### Author Response · Authors · 2024-11-22
> > **Response to Reviewer MXNH (continuation)**
> >
> > (Ending of the answer to question (3).)
> >
> > However, to further showcase the performance of our solver in the case of large data space dimensions, we conducted a **new experiment** for estimating the barycenter of images of young and elderly individuals from the FFHQ (Karras et al., 2019) dataset. Here we run our solver in the latent space of the pretrained StyleGAN2-ada (Karras et al., 2020) generator consisting of $18\times 512$ vectors, i.e., having 18 times bigger dimension than we considered before. This experiment highlights the potential of our approach to manipulate images by learning barycenters between distinct image distributions, enabling controlled transitions across semantic attributes.
> >
> > *More detailed explanation regarding this **new experiment** is given in Appendix C.2 in the revised version of our paper.*
> >
> >
> > **(4) Why don't you compare the methods (Nietert et al., 2022, Wang et al., 2024) based on robust OT? Please add corresponding experiments.**
> >
> > Thank you for your question. However, we humbly think that comparing with these methods is out of scope of our paper. To our understanding, the work (Nietert et. al., 2022) (we added the citation to the new revision) proposes a notion of robust Wasserstein distance (alternative to what we use). They do not indicate any way to use their distance in the context of barycenter problem, i.e., they do not propose any barycenter solver. Moreover, the proposed method does not learn the (robust) OT mapping itself - practically, their objective is a modification of WGAN loss (which operates only with discirminator or critic), it does not recover the desired mapping between the source and target distributions.  Therefore, it is **impossible to compare** with them (even like in our experimental Section 5.1).
> >
> > At the same time, exploring the possibility to adapt their robust Wasserstein distance for the barycenter problem is definitely an interesting point for future research, but it will require a separate research with its own theory and experimental validation. Regarding the work (Wang et. al., 2024) (mentined in Related works section 3) - they propose a *discrete* robust Wasserstein barycenter solver which falls out of our considered *continuous* computational setup, see our Section 2.3. However, for completeness, we tested the performance of the classic discrete approach (Cuturi \& Doucet et al., 2014) for balanced OT barycenter computation in the **new experiment** on continuous barycenter estimation, see Appendix C.1 in the revised version of our paper. This experiment shows that even in the balanced barycenter case, the discrete approach provides poor approximation for the ground-truth continuous barycenter and the quality of this approximation decreases drastically with the increase of dimension.
> >
> > **Concluding remarks**. Please respond to our post to let us know if the clarifications above suitably address your concerns about our work. We are happy to address any remaining points during the discussion phase; if the responses above are sufficient, we kindly ask that you consider raising your score.
> >
> > **References.**
> >
> > J. Choi, J. Choi, and M. Kang. Generative modeling through the semi-dual formulation of unbalanced optimal transport. In Advances in Neural Information Processing Systems, volume 36, 2023.
> >
> > J. Choi, J. Choi, and M. Kang. Analyzing and Improving Optimal-Transport-based Adversarial Networks. International Conference on Learning Representations, 2024.
> >
> > Marco Cuturi and Arnaud Doucet. Fast computation of wasserstein barycenters. In International conference on machine learning. PMLR, 2014
> >
> > Nietert S, Goldfeld Z, Cummings R. Outlier-robust optimal transport: Duality, structure, and statistical analysis[C]. International Conference on Artificial Intelligence and Statistics. PMLR, 2022.
> >
> > Wang X, Huang J, Yang Q, et al. On Robust Wasserstein Barycenter: The Model and Algorithm[C]. Proceedings of the 2024 SIAM International Conference on Data Mining (SDM), 2024
> >
> > Pedro C Álvarez-Esteban, E Del Barrio, JA Cuesta-Albertos, and C Matrán. A fixed-point approach to barycenters in wasserstein space. Journal of Mathematical Analysis and Applications, 2016.
> >
> > Kolesov et. al., Estimating Barycenters of Distributions with Neural Optimal Transport, Proceedings of the 41st International Conference on Machine Learning, 2024a
> >
> > Kolesov et. al., Energy-Guided Continuous Entropic Barycenter Estimation for General Costs, NeurIPS, 2024b
> >
> > Tero Karras, Samuli Laine, and Timo Aila. A style-based generator architecture for generative adversarial networks. CVPR, 2019
> >
> > Korotin, A., Li, L., Genevay, A., Solomon, J. M., Filippov, A., & Burnaev, E. (2021). Do neural optimal transport solvers work? a continuous wasserstein-2 benchmark. NeurIPS, 34.
> >
> >
> > Nguyen, N. H., Le, D., Nguyen, H. P., Pham, T., & Ho, N. (2024). On Barycenter Computation: Semi-Unbalanced Optimal Transport-based Method on Gaussians. arXiv preprint arXiv:2410.08117.

---

> > > ### Comment · Reviewer_MXNH · 2024-11-25
> > >
> > > Thank you for your detailed answer. I will raise your score.

---

> > > > ### Author Response · Authors · 2024-11-28
> > > > **Additional comment for Reviewer MXNH**
> > > >
> > > > Dear Reviewer,
> > > >
> > > > we appreciate your positive feedback on our answers. Please note that we have included in the revised version of our paper an additional theoretical result showing the quality bounds for the recovered plans (see our new [comment](https://openreview.net/forum?id=CI5Cj0vktS&noteId=QiKYPcKq1s) for all reviewers).
> > > >
> > > > In the previous message you mentioned that you plan to update the score. However, we see that you have not updated it yet. Are there any other questions that you would like to ask us before finalizing your score?
> > > >
> > > > Best regards,
> > > > the Authors

---

> > > > > ### Comment · Reviewer_MXNH · 2024-11-30
> > > > >
> > > > > Thank you for the reminder. There might have been an issue with the network at that time, which caused the update to fail. The update has now been successfully completed. I apologize for any inconvenience caused.

---

### Official Review · Reviewer_nQfu · 2024-11-04

**Soundness:** 2
**Presentation:** 2
**Contribution:** 1
**Rating:** 6
**Confidence:** 3

**Summary:**

This paper proposes a neural network-based method to estimate continuous barycenter via the dual formulation of the semi-unbalanced OT.

**Strengths:**

1. The proposed method may be the first continuous robust barycenter estimation approach with proper theoretical support and practical validation.

2. The proposed method is robust and can estimate the barycenter based on the data containing the outliers.

**Weaknesses:**

My main concern is the novelty of the paper. The primary method seems to be a combination of barycenter calculation, neural optimal transport, and semi-unbalanced optimal transport. In my view, simply combining methods may not be sufficient for publication at ICLR, so I give a negative score. I will consider raising my score if the authors can demonstrate unique contributions that go beyond a straightforward combination of these methods.

**Questions:**

Q1. How does the time efficiency of U-NOTB compare with other baselines?

Q2. The experiments seem to lack details about the size of the training/testing sets and some training specifics, such as learning rate and other hyperparameters.

Q3. Since the overall training is similar to GANs with a max-min approach, is the training of U-NOTB stable?

---

> ### Author Response · Authors · 2024-11-22
> **Response to Reviewer nQfu**
>
> Thank you for your detailed feedback. Please find the answers to your questions below.
>
> **(1) My main concern is the novelty of the paper. <...> if the authors can demonstrate unique contributions that go beyond a straightforward combination of these methods.**
>
> We would like to clarify the novelty of our paper. It would be most correct to position our method as a generalization of the recent SOTA approach to estimating continuous **balanced** OT barycenters - **NOTB** (Kolesov et al., 2024, Estimating), to the case of **semi-unbalanced OT** barycenters. However, this generalization is not straightforward as it is based on different principles.
>
> *First*, the congruence condition on the potentials, which is used in the NOTB approach, ceases to be true in the case of semi-unbalanced OT. In our paper, we developed a completely different condition on the potentials, which required the usage of an additional parameter $m$ in the optimization algorithm.
>
> *Second*, the procedure of sampling from the barycenter during the inference has major differences from the one used in NOTB. In the semi-unbalanced case, to get samples from the barycenter, we first need to define the procedure of sampling from the left marginals of the plans. Defining such a sampling procedure is tricky even for the continuous unbalanced OT (UOT) solver by its own. As far as we know, only one of the existing continuous UOT solvers (Gazdieva et al., 2024) propose such a strategy using the **restricted** Gaussian mixture parametrization of potentials. Other scalable and theoretically justified continuous UOT solvers (Choi et al., 2024, Yang et al., 2018) do not offer such a precise sampling procedure and consider only sampling from the input measure. In contrast to previous approaches, we suggest a precise sampling procedure that utilizes equation (11) from our paper.
> **Using the learned potentials, our solver allows for precise sampling of barycenter from the left marginals of learned plans without any restrictions on the underlying potentials.**
> In the case of previous works in the field such as NOTB, the situation is much easier since these marginals coincide with input measures and sampling is defined a priori.
>
> *Third*, during the rebuttal period, we conducted **a novel high-dimensional experiment** to further demonstrate the practical advantages of our method. Specifically, as detailed in Appendix C.3, we performed interpolation between two image distributions within the FFHQ 256×256 dataset, focusing on transitioning from the *young* to *elderly* categories. This experiment highlights the potential of our approach to manipulate images by learning barycenters between distinct image distributions, enabling controlled transitions across semantic attributes. Moreover, our model also demonstrates robustness to class imbalancedness in this practical task compared to other baselines.
>
> *More detailed explanation regarding this experiment is given in Appendix C.3 in the revised version of our paper.*
>
> **(2) How does the time efficiency of U-NOTB compare with other baselines?**
>
> In the revised manuscript, we reported the training/inference time of our method, see **new Tables 3,4** in Appendix B.1. As shown in Table 3, the time efficiency of U-NOTB is comparable to SOTA balanced solver NOTB in terms of training time. In terms of inference time, U-NOTB is slower, taking 2 to 10 times longer than NOTB. This gap is due to the additional computational complexity introduced by rejection sampling, which requires calculating the $c$-transform of the potential function $\hat{f}_k$. This process involves a forward pass through the learned potential function. However, we would like to highlight that thanks to our proposed sampling method, our solver gains *robustness to outliers* and *class imbalance*. A detailed discussion of rejection sampling can be found in lines 333–338 of our manuscript.
>
> **(3) The experiments seem to lack details about the size of the training/testing sets and some training specifics, such as learning rate and other hyperparameters.**
>
> The implementation details for our solver are given in Appendix B. Specifically, it includes Table 2 with all types of hyperparameters which we used in the experiments. We refer to this Appendix section in the main text of our paper, see lines 346-347. However, we would be happy to include the additional details in a revised version of our paper if you can point out which information is missing. *Could you please clarify which extra details should be included in our paper?*

---

> ### Author Response · Authors · 2024-11-22
> **Response to Reviewer nQfu (continuation)**
>
> **(4) Since the overall training is similar to GANs with a max-min approach, is the training of U-NOTB stable?**
>
> Thanks for raising this point. Indeed, the instability of training
> is a well-known issue of adversarial optimization objectives. Our approach is not an exception and we noted this aspect in Discussion section of our paper, see lines 534-537. Still, we emphasize that the majority of the existing approaches for barycenter computation suffer from some kind of computational issues. Among the recent approaches: (Kolesov et. al., 2024, Estimating), (Korotin et. al., 2022) also resort to min-max optimization; (Kolesov et. al., 2024, Energy) utilizes Energy-based training with Langevin simulation which is time costly and may be unstable under improper hyperparameters setting (e.g., too large Langevin step size, too small number of Langevin steps, etc.). Among the other approaches: while there may be some that do not present serious computational challenges, this comes at the cost of limited applicability (e.g., reliance exclusively to $\ell_2$ OT) and scalability, see Table 1 in (Kolesov et. al., 2024, Energy).
>
>
> **Concluding remarks**. Please respond to our post to let us know if the clarifications above suitably address your concerns about our work. We are happy to address any remaining points during the discussion phase; if the responses above are sufficient, we kindly ask that you consider raising your score.
>
> **References.**
>
> J. Choi, J. Choi, and M. Kang. Generative modeling through the semi-dual formulation of unbalanced optimal transport. In Advances in Neural Information Processing Systems, volume 36, 2024.
>
> M. Gazdieva, A. Asadulaev, E. Burnaev, and A. Korotin. Light Unbalanced Optimal Transport. In Advances in Neural Information Processing Systems, volume 36, 2024.
>
> Kolesov et. al., Estimating Barycenters of Distributions with Neural Optimal Transport, Proceedings of the 41st International Conference on Machine Learning, PMLR 235:25016-25041, 2024.
>
> Korotin et al. Wasserstein iterative networks for barycenter estimation. Advances in Neural Information Processing Systems 35, 15672-15686, 2022.
>
> Kolesov et. al., Energy-Guided Continuous Entropic Barycenter Estimation for General Costs, NeurIPS, 2024.
>
> K. D. Yang and C. Uhler. Scalable unbalanced optimal transport using generative adversarial networks. In International Conference on Learning Representations, 2018.

---

> > ### Author Response · Authors · 2024-11-28
> > **Additional comment for Reviewer nQfu**
> >
> > Dear Reviewer,
> >
> > The deadline for the rebuttal phase is fast approaching. We would be grateful if you could provide us with your feedback on our responses to the reviews. We are happy to address any additional points during the remaining period.
> >
> > Best regards,
> > the Authors

---

> > > ### Author Response · Authors · 2024-12-02
> > >
> > > Dear reviewer,
> > >
> > > As the rebuttal phase deadline is approaching in a few hours, we would greatly appreciate your feedback on our responses to the reviews.
> > >
> > > Thank you.
> > >
> > > Best regards, The Authors

---

> > > > ### Comment · Reviewer_nQfu · 2024-12-03
> > > >
> > > > Thank you for your response. I will increase my score.

---

### Author Response · Authors · 2024-11-22
**General response**

Dear reviewers,

thank you for your thoughtful reviews! We appreciate that you positively highlight our theoretical insights (Reviewers nQfu, MXHN), proper practical validation (Reviewers nQfu, niQu) and overall quality of the text (Reviewer hyxj).

We have uploaded an updated version of the paper. The newly added content is highlighted with the **blue** color. **The changes include**:

- [nQfu] New Table with training/inference times of our solver and baselines in **Appendix B.1**;
- [MXNH] New **Appendicies C.1/C.2** showing the performance of our solver and baselines in OT/SUOT barycenter problem for Gaussian distributions with known ground-truth solutions. The experiments reveal the poor performance of classic discrete method (Cuturi & Doucet, 2014) in continuous OT barycenter problem;
- [MXHN] New **Appendix C.3** with high-dimensional experiment demonstrating the practical advantages of our solver;
- [MXHN, niQu] Added references to papers (Nietert et al., 2022), (Wang et al., 2024), (Séjourné et al., 2023), see **Section 3** and **Section 6**;
- [niQu] Fixed the unclear aspects/typos in the **main text**.

**References.**

Marco Cuturi and Arnaud Doucet. Fast computation of wasserstein barycenters. In International
conference on machine learning, pp. 685–693. PMLR, 2014

Nietert S, Goldfeld Z, Cummings R. Outlier-robust optimal transport: Duality, structure, and statistical analysis[C]. International Conference on Artificial Intelligence and Statistics. PMLR, 2022: 11691-11719.

Wang X, Huang J, Yang Q, et al. On Robust Wasserstein Barycenter: The Model and Algorithm[C]. Proceedings of the 2024 SIAM International Conference on Data Mining (SDM). Society for Industrial and Applied Mathematics, 2024: 235-243

Séjourné, T., Bonet, C., Fatras, K., Nadjahi, K., \& Courty, N. (2023). Unbalanced optimal transport meets sliced-Wasserstein. arXiv preprint arXiv:2306.07176.

---

### Author Response · Authors · 2024-11-28
**New theoretical result**

Dear Reviewers,

as per the request of the Reviewer hyxj, we have prepared an additional revision of our paper which includes a **new theoretical result** (Theorem 3 in Section 4) establishing the **quality bounds** on the recovered plans based on the duality gaps, i.e., the errors for solving inner and outer optimization problems in our objective (9). Theorem 3 shows that when our Algorithm 1 optimizing this $\max$-$\min$ objective converged nearly to the optimum, its solutions are close to the true conditional plans.

Best regards,
the Authors

---

### Meta-Review · Area_Chair_Q999 · 2024-12-20

**Metareview:**

In the paper, the authors proposed a new approach to estimate the (semi)-unbalanced barycenter of continuous distributions. It is done via formulated the problem as a min-max optimization problem and the method can be used for any general cost function. All the reviewers agree that the proposed method is novel and the theoretical results are thorough. During the rebuttal period, the authors also included additional theoretical results on the quality bounds on the recovered plans based on the duality gaps, which strengthens the theoretical guarantee for the Algorithm 1 in the paper.

While there are some concerns about the experiments that are not extensive as well as the method may be slightly incremental compared to the previous works, I believe that the work has sufficient novelty and merit for ICLR. Therefore, I recommend accepting the paper.

The authors are encouraged to incorporate the suggestions of the reviewers into the revision of their manuscript.

**Additional Comments On Reviewer Discussion:**

Please refer to the meta-review.

---

### Decision · Program_Chairs · 2025-01-22

Accept (Poster)